# Deep Learning for Continuous-Time Stochastic Control with Jumps

**Patrick Cheridito**
Department of Mathematics
ETH Zurich, Switzerland

**Jean-Loup Dupret**[*]
Department of Mathematics
ETH Zurich, Switzerland

**Donatien Hainaut**
LIDAM-ISBA
UCLouvain, Belgium

{patrickc,jdupret}@ethz.ch, {donatien.hainaut}@uclouvain.be

## Abstract

In this paper, we introduce a model-based deep-learning approach to solve finite-horizon continuous-time stochastic control problems with jumps. We iteratively train two neural networks: one to represent the optimal policy and the other to approximate the value function. Leveraging a continuous-time version of the dynamic programming principle, we derive two different training objectives based on the Hamilton–Jacobi–Bellman equation, ensuring that the networks capture the underlying stochastic dynamics. Empirical evaluations on different problems illustrate the accuracy and scalability of our approach, demonstrating its effectiveness in solving complex high-dimensional stochastic control tasks. Code is available at https://github.com/jdupret97/Deep-Learning-for-CT-Stochastic-Control-with-Jumps.

## 1 Introduction

A large class of dynamic decision-making problems under uncertainty can be modeled as continuous-time stochastic control problems. In this paper, we introduce two neural network-based numerical algorithms for such problems in high dimensions with finite time horizon and jumps. More precisely, we consider control problems of the form

$$\sup_{\alpha} \mathbb{E}\left[\int_0^T f(t, X_t^{\alpha}, \alpha_t)dt + F(X_T^{\alpha})\right], \tag{1}$$

for a finite horizon $T > 0$, where the supremum is over predictable control processes $\alpha = (\alpha_t)_{0 \le t \le T}$ taking values in a subset $A \subseteq \mathbb{R}^m$. The controlled process $X^{\alpha}$ evolves in a subset $D \subseteq \mathbb{R}^d$ according to

$$dX_t^{\alpha} = \beta(t, X_t^{\alpha}, \alpha_t)dt + \sigma(t, X_t^{\alpha}, \alpha_t)dW_t + \int_E \gamma(t, X_{t-}^{\alpha}, z, \alpha_t)N^{\alpha}(dz, dt), \ \ X_0^{\alpha} = x \in D, \tag{2}$$

for an initial condition $x \in D$, an $n$-dimensional Brownian motion $W$ and a controlled random measure $N^{\alpha}$ on $E \times \mathbb{R}_+$, with $E = \mathbb{R}^l \setminus \{0\}$, and suitable functions $\beta \colon [0, T] \times D \times A \to \mathbb{R}^d$, $\sigma \colon [0, T] \times D \times A \to \mathbb{R}^{d \times n}$ and $\gamma \colon [0, T] \times D \times E \times A \to \mathbb{R}^d$. The functions $f \colon [0, T] \times D \times A \to \mathbb{R}$ and $F \colon D \to \mathbb{R}$ model the running and final rewards, respectively. We assume the controlled random measure is given by $N^{\alpha}(B \times [0, t]) = \sum_{j=1}^{M_t^{\alpha}} 1_{\{Z_j \in B\}}$ for measurable subsets $B \subseteq E$, where $M^{\alpha}$ is a Cox process with a stochastic intensity of the form $\lambda(t, X_{t-}^{\alpha}, \alpha_t)$ and $Z_1, Z_2, \ldots$ are i.i.d. $E$-valued random vectors such that, conditionally on $\alpha$, the random elements $W, M^{\alpha}$ and $Z_1, Z_2, \ldots$ are independent. Our goal is to find an optimal control $\alpha^*$ and the corresponding value of problem

---

[*]Corresponding author.

39th Conference on Neural Information Processing Systems (NeurIPS 2025).

(1). In view of the Markovian nature of the dynamics (2), we work with feedback controls of the form $\alpha_t = \alpha(t, X_{t-}^\alpha)$ for measurable functions $\alpha \colon [0, T) \times D \to A$ and consider the value function $V \colon [0, T] \times D \to \mathbb{R}$ given by

$$V(t, x) = \sup_\alpha \mathbb{E}\left[ \int_t^T f(s, X_s^\alpha, \alpha_s) ds + F(X_T^\alpha) \,\Big|\, X_t^\alpha = x \right].$$

Under suitable assumptions[2], $V$ is the unique solution of the following Hamilton–Jacobi–Bellman (HJB) equation

$$\partial_t V(t, x) + \sup_{a \in A} H(t, x, V, a) = 0, \quad V(T, x) = F(x), \tag{3}$$

for the Hamiltonian $H : [0, T) \times D \times \mathcal{V} \times A \to \mathbb{R}$ given[3] by

$$\begin{aligned}
H(t, x, V, a) := f(t, x, a) + \beta^T(t, x, a) \nabla_x V(t, x) + \frac{1}{2} \mathrm{Tr}\big[ \sigma\sigma^T(t, x, a) \nabla_x^2 V(t, x) \big] \\
+ \lambda(t, x, a) \mathbb{E}[V(t, x + \gamma(t, x, Z_1, a)) - V(t, x)],
\end{aligned} \tag{4}$$

where $\nabla_x V$, $\nabla_x^2 V$ denote the gradient and Hessian of $V$ with respect to $x$. Our approach consists in iteratively training two neural networks to approximate the value function $V$ and an optimal control $\alpha^*$ attaining $V$. It has the following features:

- It yields accurate results for high-dimensional continuous-time stochastic control problems in cases where the underlying system dynamics are known.
- It can effectively handle a combination of diffusive noise and random jumps with controlled intensities.
- It can handle general situations where the optimal control is not available in closed form but has to be learned together with the value function.
- It approximates both the value function and optimal control at any point $(t, x) \in [0, T) \times D$.

While methods like finite differences, finite elements and spectral methods work well for solving partial (integro-)differential equations P(I)DEs in low dimensions, they suffer from the curse of dimensionality and, as a consequence, become infeasible in high dimensions. Recently, different deep learning based approaches for solving high-dimensional PDEs have been proposed (Raissi et al., 2017, 2019; Han et al., 2017, 2018; Sirignano & Spiliopoulos, 2018; Berg & Nyström, 2018; Beck et al., 2021; Lu et al., 2021; Bruna et al., 2024). They can directly be used to solve continuous-time stochastic control problems that admit an explicit solution for the optimal control in terms of the value function since in this case, the expression for the optimal control can be plugged into the HJB equation, which then reduces to a parabolic PDE. On the other hand, if the optimal control is not available in closed form, it cannot be plugged into the HJB equation, but instead, has to be approximated numerically while at the same time solving a parabolic PDE. Such implicit optimal control problems can no longer be solved directly with one of the deep learning methods mentioned above but require a specifically designed iterative approximation procedure. For low-dimensional problems with non-explicit optimal controls, a standard approach from the reinforcement learning (RL) literature is to use generalized policy iteration (GPI), a class of iterative algorithms that simultaneously approximate the value function and optimal control (Jacka & Mijatović, 2017; Sutton & Barto, 2018). Particularly popular are actor-critic methods going back to Werbos (1992), which have a separate memory structure to represent the optimal control independently of the value function. However, classical GPI schemes become impractical in high dimensions as the PDE and optimization problem both have to be discretized and solved for every point in a finite grid. This raises the need for meshfree methods to solve implicit continuous-time stochastic control problems in high dimensions with continuous action space. Several local approaches based on a time-discretization of (2) have been explored by e.g. Han & E (2016), Nüsken & Richter (2021), Huré et al. (2021), Bachouch et al. (2022), Ji et al. (2022), Li et al. (2024), Domingo-Enrich et al. (2024a), Domingo-Enrich et al. (2024b). Alternatively, (deep) RL methods can be used, for instance, Q-learning type algorithms such as DQN (Mnih et al., 2013) or C51 (Bellemare et al., 2017), see also Wang et al. (2020), Jia & Zhou (2023), Gao et al. (2024), Szpruch et al. (2024); or actor critic approaches such as DDPG (Lillicrap et al., 2019), SAC (Haarnoja

---

[2] see e.g. Soner (1988)

[3] By $\mathcal{V}$ we denote the set of all functions in $C^{1,2}([0, T) \times \mathbb{R}^d)$ such that the expectation in (4) is finite for all $t, x$ and $a$.

et al., 2018), A2C/A3C (Mnih et al., 2016), PPO (Schulman et al., 2017) or TRPO (Schulman et al., 2015). However, these RL algorithms are model-free, that is, they do not explicitly take into account the underlying dynamics (2) of the control problem (1) but instead, solely rely on sampling from the environment. As a result, they are less accurate in cases where the system dynamics are known (see Figure 3 below). On the other hand, the difficulty of solving high-dimensional PDEs is further exacerbated if the system dynamics includes stochastic jumps since in this case, the HJB equation (3) requires the computation of the jump expectations $\mathbb{E}\,V(t, x + \gamma(t, x, Z_1, a))$ for every space-time point $(t, x)$ sampled from the domain.

In this paper, we introduce a deep model-based approach for stochastic control problems with jumps that takes the system dynamics (2) into account by leveraging the HJB equation (3). This removes the need to simulate the underlying jump-diffusion (2) and, as a result, avoids discretization errors. Our approach combines GPI with a PIDE solving method. It approximates the value function and optimal control in an actor-critic fashion with two neural networks trained iteratively on sampled data from the space-time domain. As such, it has the advantage that it provides global approximations of the value function and optimal control available for all space-time points, which can be evaluated rapidly in online applications. We develop two related algorithms. The first one, GPI-PINN, approximates the value function by training a neural network to minimize the residuals of the HJB equation (3), following a physics-inspired neural network (PINN) approach (Raissi et al., 2017, 2019) while leveraging Proposition 3.1 below to avoid the direct computation of the gradient $\nabla_x V$ and Hessian $\nabla_x^2 V$ in the Hamiltonian (4). GPI-PINN can be viewed as an extension of the method proposed by Duarte et al. (2024) adapted to a finite-horizon setup with time-dependence and a terminal condition in the HJB equation, leading to time-dependent optimal control strategies and value function, see also Dupret & Hainaut (2024). It works well in high dimensions for control problems without jumps in the underlying dynamics (2) ($\gamma = 0$), but becomes inefficient in the presence of jumps as it requires the computation of the jump-expectation $\mathbb{E}\,V(t, x + \gamma(t, x, Z_1, a))$ at numerous sample points $(t, x)$ in every iteration of the algorithm. To address this, our second algorithm, GPI-CBU, relies on a continuous-time Bellman updating rule to approximate the value function, thereby circumventing the computation of gradients, Hessians and jump-expectations altogether. This makes it highly efficient for high-dimensional stochastic control problems with jumps, even when the control is not available in closed form.

We illustrate the accuracy and scalability of our approach in different numerical examples and provide comparisons with popular RL and deep-learning control methods. Proofs of theoretical results and additional numerical experiments are given in the Appendix.

## 2  General approach

Let $\alpha \colon [0, T] \times D \to A$ be a feedback control such that equation (2) has a unique solution $X^\alpha$ and consider the corresponding value function

$$V^\alpha(t, x) = \mathbb{E}\left[ \int_t^T f(s, X_s^\alpha, \alpha(s, X_{s-}^\alpha))ds + F(X_T^\alpha) \,\middle|\, X_t^\alpha = x \right], \quad (t, x) \in [0, T] \times D.$$

Under appropriate assumptions, one obtains the following two results from standard arguments[4].

**Theorem 2.1** (Feynman–Kac Formula). *$V^\alpha$ satisfies the PIDE*

$$\partial_t V^\alpha(t, x) + H(t, x, V^\alpha, \alpha(t, x)) = 0, \quad V^\alpha(T, x) = F(x).$$

**Theorem 2.2** (Verification Theorem). *Let $v \in \mathcal{V} \cap C([0, T] \times D)$ be a solution of the HJB equation (3) such that there exists a measurable mapping $\hat{\alpha} : [0, T] \times D \to A$ satisfying*

$$\hat{\alpha}(t, x) \in \arg\max_{a \in A} H(t, x, v, a) \quad \text{for all } (t, x) \in [0, T] \times D$$

*and the controlled jump-diffusion equation (2) admits a unique solution for each initial condition $x \in D$. Then $v = V$ and $\hat{\alpha}$ is an optimal control.*

Based on Theorems 2.1 and 2.2, we iteratively approximate the value function $V$ and optimal control $\alpha^*$ with neural networks[5] $V_\theta \colon [0, T] \times D \to \mathbb{R}$ and $\alpha_\phi \colon [0, T] \times D \to A$. For given $\alpha_\phi$, we train $V_\theta$

---

[4]see e.g. the arguments in the proofs of Theorems 1.3.1 and 2.2.4 in Bouchard (2021).

[5]Using a $C^2$-activation function in the network $V_\theta$ ensures that it belongs to $C^2([0, T] \times \mathbb{R}^d)$.

so as to solve the controlled HJB equation

$$\partial_t V_\theta(t,x) + H(t,x,V_\theta,\alpha_\phi(t,x)) = 0, \quad V_\theta(T,x) = F(x), \tag{5}$$

while for given $V_\theta$, $\alpha_\phi$ is trained with the goal to maximize the Hamiltonian $H(t,x,V_\theta,\alpha_\phi(t,x))$.

In the following, we introduce two different training objectives for updating the value network $V_\theta$, leading respectively to the algorithms GPI-PINN and GPI-CBU. GPI-PINN uses a PINN-type loss together with a trick adapted from Duarte et al. (2024) to bypass the explicit computation of gradients and Hessians, whereas GPI-CBU relies on a continuous-time Bellman updating rule with an expectation-free version of the Hamiltonian, thereby also avoiding the computation of the jump-expectations in (4).

## 3  GPI-PINN

GPI-PINN, described in Algorithm 1 below, relies on a PINN approach to minimize the residuals of the controlled HJB equation (5) in the value function approximation step. To avoid explicit computations of the gradient $\nabla_x V_\theta(t,x)$ and Hessian $\nabla_x^2 V_\theta(t,x)$, which appear in the Hamiltonian (4), we use the following trick, adapted from Duarte et al. (2024).

**Proposition 3.1.** *Consider a function $v \in \mathcal{V}$ together with a pair $(t,x) \in [0,T] \times D$. Define the function $\psi : \mathbb{R} \to \mathbb{R}$ by*

$$\psi(h) := \sum_{i=1}^{n} v\left(t + \frac{h^2}{2n}, x + \frac{h}{\sqrt{2}}\sigma_i(t,x,a) + \frac{h^2}{2n}\beta(t,x,a)\right),$$

*where $\sigma_i(t,x,a)$ is the $i^{th}$ column of the $d \times n$ matrix $\sigma(t,x,a)$. Then,*

$$\psi''(0) = \partial_t v(t,x) + \beta^\top(t,x,a)\,\nabla_x v(t,x) + \frac{1}{2}\mathrm{Tr}\left[\sigma\sigma^\top(t,x,a)\nabla_x^2 v(t,x)\right].$$

Proposition 3.1 makes it possible to replace the computation of gradients and Hessians of $v$ by evaluating the univariate function $\psi''(0)$, the cost of which, using automatic differentiation, is a small multiple of $n \cdot cost(v)$.

To formulate GPI-PINN, we need the extended Hamiltonian

$$\mathcal{H}(t,x,v,a) := \partial_t v(t,x) + H(t,x,v,a), \tag{6}$$

which by Proposition 3.1, can be written as

$$\mathcal{H}(t,x,v,a) = \psi''(0) + f(t,x,a) + \lambda(t,x,a)\mathbb{E}\big[v(t,x+\gamma(t,x,Z_1,a)) - v(t,x)\big].$$

In Algorithm 1, we simplify the notation by using $\mathcal{H}(t,x,\theta,\phi) := \mathcal{H}(t,x,V_\theta,\alpha_\phi(t,x))$.

The loss function $\mathscr{L}_1$ in (7) consists of two terms. The first represents the expected PIDE residual in the interior of the space-time domain with respect to a suitable measure $\mu$ on $[0,T] \times D$. The second term penalizes violations of the terminal condition according to a measure $\nu$ on $D$. Hence, $\mathscr{L}_1$ measures how well the function $V_\theta$ satisfies the controlled HJB equation (5) corresponding to a control $\alpha_\phi$. In every epoch $k$, the goal in Step 1 is therefore to find a parameter vector $\theta$ such that the value network $V_\theta$ minimizes the error $\mathscr{L}_1(\theta, \phi^{(k)})$. We do this with a mini-batch stochastic gradient method which updates the measures $\mu$ and $\nu$ according to the residual-based adaptive distribution (RAD) method of Wu et al. (2023), as it is known to significantly improve the accuracy of PINNs. In Step 2, we minimize $\mathscr{L}_2(\theta^{(k+1)}, \phi)$ with respect to $\phi$. This corresponds to choosing the control $\alpha_\phi$ so as to maximize the extended Hamiltonian $\mathcal{H}$ (or equivalently $H$); see Duan et al. (2023), Dupret & Hainaut (2024) and Cohen et al. (2025) for theoretical convergence results supporting this approach.

Since the expectations in $\mathscr{L}_1$, $\mathscr{L}_2$ and the extended Hamiltonian (6) are typically not available in closed form, we replace them by sample-based estimates. First, we estimate $\mathcal{H}(t,x,\theta,\phi)$ with $\widehat{\mathcal{H}}(t,x,\theta,\phi)$ by approximating the jump-expectation

$$\mathbb{E}\,V_\theta(t,x+\gamma(t,x,Z_1,a_\phi(t,x))) \approx \frac{1}{J}\sum_{j=1}^{J} V_\theta(t,x+\gamma(t,x,z_j,a_\phi(t,x))) \tag{9}$$

---

**Algorithm 1** GPI-PINN

---

Initialize admissible weights $\theta^{(0)}$ for $V_\theta$ and $\phi^{(0)}$ for $\alpha_\phi$. Choose proportionality factors $\xi_1, \xi_2 > 0$ and set epoch $k = 0$.
**repeat**

   **Step 1:** Update the value network $V_{\theta^{(k+1)}}$ for a given control $\alpha_{\phi^{(k)}}$ by minimizing the loss

$$\theta^{(k+1)} = \arg\min_\theta \mathscr{L}_1(\theta, \phi^{(k)}),$$

   where

$$\mathscr{L}_1(\theta, \phi) = \xi_1 \, \mathbb{E}_{(t,x)\sim\mu} \mathcal{H}^2(t, x, \theta, \phi) + \xi_2 \, \mathbb{E}_{x\sim\nu} \left(V_\theta(T, x) - F(x)\right)^2 \qquad (7)$$

   **Step 2:** Update the control network $\alpha_{\phi^{(k+1)}}$ for a given value network $V_{\theta^{(k+1)}}$ by minimizing the loss

$$\phi^{(k+1)} = \arg\min_\phi \mathscr{L}_2(\theta^{(k+1)}, \phi),$$

   where

$$\mathscr{L}_2(\theta, \phi) = -\mathbb{E}_{(t,x)\sim\mu} \mathcal{H}(t, x, \theta, \phi) \qquad (8)$$

$k \leftarrow k + 1$
**until** some convergence criterion is satisfied.
**return** $V_{\theta^{(k)}}$ and $\alpha_{\phi^{(k)}}$ and set $k_* \leftarrow k$.

---

for points $(z_j)_{j=1}^J$ in $E$ sampled from the distribution $\mathcal{Z}$ of $Z_1$. Then, we approximate $\mathscr{L}_1$ and $\mathscr{L}_2$ in every time step by

$$\widehat{\mathscr{L}_1}(\theta, \phi^{(k)}) = \frac{\xi_1}{M_1} \sum_{m=1}^{M_1} \left(\widehat{\mathcal{H}}(t_m, x_m, \theta, \phi^{(k)})\right)^2 + \frac{\xi_2}{M_2} \sum_{m=1}^{M_2} \left(V_\theta(T, y_m) - F(y_m)\right)^2$$

and

$$\widehat{\mathscr{L}_2}(\theta^{(k+1)}, \phi) = -\frac{1}{M_1} \sum_{m=1}^{M_1} \widehat{\mathcal{H}}(t_m, x_m, \theta^{(k+1)}, \phi),$$

where $(t_m, x_m)_{m=1}^{M_1} \in [0, T) \times D$ and $(y_m)_{m=1}^{M_2} \in D$ are sampled from $\mu$ and $\nu$, respectively. In every epoch $k = 0, \ldots, k_* - 1$, we initialize $\theta_0^{(k)} := \theta^{(k)}$ and make $N_1$ gradient steps

$$\theta_{i+1}^{(k)} = \theta_i^{(k)} - \eta_1 \nabla_\theta \widehat{\mathscr{L}_1}(\theta_i^{(k)}, \phi^{(k)}) = \theta_i^{(k)} - \frac{2\eta_1 \xi_1}{M_1} \sum_{m=1}^{M_1} \widehat{\mathcal{H}}(t_m, x_m, \theta_i^{(k)}, \phi^{(k)}) \nabla_\theta \widehat{\mathcal{H}}(t_m, x_m, \theta_i^{(k)}, \phi^{(k)})$$

$$- \frac{2\eta_1 \xi_2}{M_2} \sum_{m=1}^{M_2} \left(V_{\theta_i^{(k)}}(T, y_m) - F(y_m)\right) \nabla_\theta V_{\theta_i^{(k)}}(T, y_m), \qquad (10)$$

$i = 0, \ldots, N_1 - 1$, to obtain $\theta^{(k+1)} = \theta_{N_1}^{(k)}$. Then, we initialize $\phi_0^{(k)} = \phi^{(k)}$ and perform $N_2$ gradient steps

$$\phi_{i+1}^{(k)} = \phi_i^{(k)} - \eta_2 \nabla_\phi \widehat{\mathscr{L}_2}(\theta^{(k+1)}, \phi_i^{(k)}) = \phi_i^{(k)} + \frac{\eta_2}{M_1} \sum_{m=1}^{M_1} \nabla_\phi \widehat{\mathcal{H}}(t_m, x_m, \theta^{(k+1)}, \phi_i^{(k)}), \qquad (11)$$

$i = 0, \ldots, N_2 - 1$, to get $\phi^{(k+1)} = \phi_{N_2}^{(k)}$.

GPI-PINN yields global approximations $V_{\theta^{(k_*)}}$ and $\alpha_{\phi^{(k_*)}}$ of the value function and optimal control on the whole space-time domain $[0, T] \times D$. By using Proposition 3.1, it avoids the computation of the gradients and Hessians appearing in the Hamiltonian (4). However, it still has two drawbacks that make it inefficient for high-dimensional control problems with jumps. First, it has to approximate the jump-expectations $\mathbb{E}\left[V_\theta(t_m, x_m + \gamma(t_m, x_m, Z_1, \alpha_\phi(t_m, x_m)))\right]$ for all sample points $(t_m, x_m)$, $m = 1, \ldots, M_1$, in each of the gradient steps (10)–(11) and all epochs $k = 0, 1, \ldots, k_*$; see (9). Secondly, since the Hamiltonian is already a second-order integro-differential operator, the gradient steps $\nabla_\theta \widehat{\mathcal{H}}$ in (10) require the computation of third order derivatives, which is numerically costly.

# 4 GPI-CBU

GPI-CBU addresses the shortcomings of GPI-PINN by using a value function updating rule based on the expectation-free operator $G_\zeta : [0,T] \times D \times E \times \mathcal{V} \times A \to \mathbb{R}$ given by

$$G_\zeta(t,x,z,v,a) := v(t,x) + \zeta \big[ \partial_t v(t,x) + f(t,x,a) + \beta^\top(t,x,a) \nabla_x v(t,x)$$
$$+ \frac{1}{2} \mathrm{Tr} \big[ \sigma\sigma^\top(t,x,a) \nabla_x^2 v(t,x) \big] + \lambda(t,x,a) \left( v(t, x + \gamma(t,x,z,a)) - v(t,x) \right) \big]$$

for a scaling factor $\zeta \in \mathbb{R}$.

**Proposition 4.1.** *Let $X^\alpha$ be a solution of the jump-diffusion equation* (2) *corresponding to a feedback control $\alpha : [0,T] \times D \to A$ with associated value function $V^\alpha \in C^{1,2}([0,T] \times D)$. For given $t \in [0,T)$, let $Y_t$ be a $D$-valued random variable independent of $Z_1$ such that $\mathbb{E}\, G_\zeta^2(t, Y_t, Z_1, V^\alpha, \alpha(t, Y_t)) < \infty$. Then $V^\alpha(t, Y_t) = g(Y_t)$ for the Borel measurable function $g : D \to \mathbb{R}$ minimizing the mean squared error*

$$\mathbb{E}\Big[ \big( g(Y_t) - G_\zeta(t, Y_t, Z_1, V^\alpha, \alpha(t, Y_t)) \big)^2 \Big].$$

Proposition 4.1 suggests to update[6] the value function parameters according to

$$\theta^{(k+1)} = \arg\min_\theta \mathbb{E} \int_0^T \left( V_\theta(t, Y_t) - G_\zeta\big(t, Y_t, Z_1, V_{\theta^{(k)}}, \alpha(t, Y_t)\big) \right)^2 dt. \tag{12}$$

By adding a penalty term enforcing the terminal condition, we obtain the recursive scheme

$$\theta^{(k+1)} = \arg\min_\theta \mathscr{L}_1^{(k)}(\theta) \tag{13}$$

for the loss

$$\mathscr{L}_1^{(k)}(\theta) = \xi_1 \mathbb{E}_{(t,x,z) \sim \mu \otimes \mathcal{Z}} \left( V_\theta(t,x) - G_\zeta\big(t,x,z,\theta^{(k)},\phi^{(k)}\big) \right)^2 + \xi_2 \mathbb{E}_{x \sim \nu} \left( V_\theta(T,x) - F(x) \right)^2,$$

where we use the notation $G_\zeta(t,x,z,\theta,\phi) := G_\zeta(t,x,z,V_\theta,\alpha_\phi(t,x))$. To implement (13), we approximate $\mathscr{L}_1^{(k)}(\theta)$ with

$$\widehat{\mathscr{L}}_1^{(k)}(\theta) = \frac{\xi_1}{M_1} \sum_{m=1}^{M_1} \left( V_\theta(t_m, x_m) - G_\zeta\big(t_m, x_m, z_m, \theta^{(k)}, \phi^{(k)}\big) \right)^2 + \frac{\xi_2}{M_2} \sum_{m=1}^{M_2} (V_\theta(T, y_m) - F(y_m))^2 \tag{14}$$

for $(t_m, x_m, z_m)_{m=1}^{M_1} \in [0,T) \times D \times E$ sampled from $\mu \otimes \mathcal{Z}$ and $(y_m)_{m=1}^{M_2} \in D$ sampled from $\nu$. In every epoch $k = 0, \ldots, k_* - 1$, we initialize $\theta_0^{(k)} = \theta^{(k)}$ and perform $N_1$ gradient steps

$$\theta_{i+1}^{(k)} = \theta_i^{(k)} - \eta_1 \nabla_\theta \widehat{\mathscr{L}}_1^{(k)}(\theta_i^{(k)}) = \theta_i^{(k)} - \frac{2\eta_1 \xi_2}{M_2} \sum_{m=1}^{M_2} \left( V_{\theta_i^{(k)}}(T, y_m) - F(y_m) \right) \nabla_\theta V_{\theta_i^{(k)}}(T, y_m)$$
$$- \frac{2\eta_1 \xi_1}{M_1} \sum_{m=1}^{M_1} \left( V_{\theta_i^{(k)}}(t_m, x_m) - G_\zeta\big(t_m, x_m, z_m, \theta_0^{(k)}, \phi^{(k)}\big) \right) \nabla_\theta V_{\theta_i^{(k)}}(t_m, x_m), \tag{15}$$

$i = 0, \ldots, N_1 - 1$, to obtain $\theta^{(k+1)} = \theta_{N_1}^{(k)}$. To update the control network $\alpha_\phi$, we introduce the operator

$$\mathcal{G}(t,x,z,\theta,\phi) := \partial_t V_\theta(t,x) + f(t,x,\alpha_\phi(t,x)) + \beta^\top(t,x,\alpha_\phi(t,x)) \nabla_x V_\theta(t,x)$$
$$+ \frac{1}{2} \mathrm{Tr} \big[ \sigma\sigma^\top(t,x,\alpha_\phi(t,x)) \nabla_x^2 V_\theta(t,x) \big] + \lambda(t,x,\alpha_\phi(t,x)) \left( V_\theta(t, x + \gamma(t,x,z,\alpha_\phi(t,x))) - V_\theta(t,x) \right),$$

which is an expectation-free version of the extended Hamiltonian $\mathcal{H}$ that, instead of the jump-expectation $\mathbb{E}\, V_\theta(t, x + \gamma(t, x, Z_1, a_\phi(t,x)))$, only contains a single jump $V_\theta(t, x + \gamma(t,x,z,a_\phi(t,x)))$, $z \in E$. GPI-CBU updates the parameters of the control network $\alpha_\phi$ according to

$$\phi^{(k+1)} = \arg\min_\phi \mathscr{L}_2^{(k+1)}(\phi)$$

---

[6]By (22), the update rule (12) corresponds to $V_{\theta^{(k+1)}} = T^\alpha V_{\theta^{(k)}} := V_{\theta^{(k)}} + \zeta \mathcal{H}(\cdot, V_{\theta^{(k)}}, \alpha(\cdot))$ for the continuous-time Bellman updating (CBU) operator $T^\alpha$ associated with the feedback control $\alpha$ and fixed point $T^\alpha V^\alpha = V^\alpha$.

for the loss

$$\mathcal{L}_2^{(k+1)}(\phi) = -\mathbb{E}_{(t,x,z)\sim\mu\otimes\mathcal{Z}}\,\mathcal{G}(t,x,z,\theta^{(k+1)},\phi) = -\mathbb{E}_{(t,x)\sim\mu}\mathcal{H}(t,x,\theta^{(k+1)},\phi).$$

We hence minimize the sample estimate

$$\widehat{\mathcal{L}}_2^{(k+1)}(\phi) = -\frac{1}{M_1}\sum_{m=1}^{M_1}\mathcal{G}(t_m,x_m,z_m,\theta^{(k+1)},\phi)\,, \tag{16}$$

by setting $\phi_0^{(k)} = \phi^{(k)}$ and making $N_2$ gradient steps

$$\phi_{i+1}^{(k)} = \phi_i^{(k)} - \eta_2\nabla_\phi\widehat{\mathcal{L}}_2^{(k+1)}(\phi_i^{(k)}) = \phi_i^{(k)} + \frac{\eta_2}{M_1}\sum_{m=1}^{M_1}\nabla_\phi\mathcal{G}(t_m,x_m,z_m,\theta^{(k+1)},\phi_i^{(k)}), \tag{17}$$

$i = 0,\ldots,N_2-1$, to obtain $\phi^{(k+1)} = \phi_{N_2}^{(k)}$.

---

**Algorithm 2** GPI-CBU

---

Initialize admissible neural weights $\theta^{(0)}$ for $V_\theta$ and $\phi^{(0)}$ for $\alpha_\phi$. Choose learning rates $\eta_1,\eta_2$, proportionality factors $\xi_1,\xi_2$ and numbers of gradient steps $N_1,N_2$. Set epoch $k=0$.
**repeat**
  **Step 0:** Generate $M_1$ sample points $(t_m,x_m,z_m) \in [0,T]\times D\times E$ from $\mu\otimes\mathcal{Z}$ and $M_2$ sample points $y_m \in D$ from $\nu$.
  **Step 1:** Update $V_{\theta^{(k+1)}}$ by minimizing the loss (14)

$$\theta^{(k+1)} = \arg\min_\theta \widehat{\mathcal{L}}_1^{(k)}(\theta)\,,$$

  with $N_1$ gradient steps (15).
  **Step 2:** Update $\alpha_{\phi^{(k+1)}}$ by minimizing the loss (16)

$$\phi^{(k+1)} = \arg\min_\phi \widehat{\mathcal{L}}_2^{(k+1)}(\phi)\,,$$

  with $N_2$ gradient steps (17).
  $k \leftarrow k+1$
**until** some convergence criterion is satisfied.
**return** $V_{\theta^{(k)}}$ and $\alpha_{\phi^{(k)}}$ and set $k_* \leftarrow k$.

---

Like GPI-PINN, GPI-CBU leverages Proposition 3.1 to bypass the computation of the gradients $\nabla_x V_\theta$ and Hessians $\nabla_x^2 V_\theta$. In addition, it avoids the costly computation of the jump-expectations (9) of $V_\theta$ at each sample point $(t_m,x_m)$. Also, it it does not have to compute third-order derivatives like (10) when updating the value network since only $\nabla_\theta V_\theta$ is needed in (15). This is a consequence of the fact that the value update rule (15) of GPI-CBU is recursive as $G_\zeta$ depends on the value weights $\theta^{(k)}$ computed in the previous epoch in contrast to the residual-based GPI-PINN. On the other hand, since GPI-PINN averages over different jumps in each update, it exhibits more stable convergence than GPI-CBU; see Figure 1 below. The proportionality factors $\xi_1,\xi_2$ and the scaling factor $\zeta$ are hyperparameters. $\xi_1$ and $\xi_2$ can be fine-tuned following Wang et al. (2022). While Proposition 4.1 holds for any scaling factor $\zeta \in \mathbb{R}$, its choice affects the numerical performance of GPI-CBU. In the numerical experiments of this paper, we set $\zeta = 1$ as it provides a good trade-off between convergence speed and accuracy of the improvements in GPI-CBU. On the other hand, negative scaling factors failed to converge to the true solution in all our experiments with exploding losses $\widehat{\mathcal{L}}_1^{(k)}$ and $\widehat{\mathcal{L}}_2^{(k)}$ after only a few epochs. Alternatively, one could consider an adaptive scaling factor $\zeta_k > 0$ depending on the epoch $k$. More details on hyperparameter fine-tuning are given in Appendix B. Finally, we note that the policy updating rule of GPI-CBU (17) is equivalent to that of GPI-PINN (11), except that it circumvents the computation of the jump-expectations in $\widehat{\mathcal{H}}$ since it uses the expectation-free operator $\mathcal{G}$.

## 5 Numerical experiments

In our numerical experiments, we use the Deep Galerkin Method (DGM) architecture of Sirignano & Spiliopoulos (2018) for both the value and optimal control networks as it has been shown to

empirically improve PINN performance. Details of the network design and hyperparameters are given in Appendix B.

## 5.1 Linear-quadratic regulator with jumps

We first consider the linear-quadratic regulator (LQR) problem with jumps

$$\inf_\alpha \mathbb{E}\left[\int_0^T c_1\|\alpha_s\|_2^2\,ds + c_2\|X_T^\alpha\|_2^2\right], \tag{18}$$

where the infimum is over $d$-dimensional predictable processes $(\alpha_t)_{0\le t\le T}$ and $X^\alpha$ is a $d$-dimensional process with controlled dynamics

$$dX_t^\alpha = \alpha_t dt + BdW_t + d\sum_{j=1}^{M_t^\alpha} Z_j\,,\quad X_0 = x \in \mathbb{R}^d \tag{19}$$

for a $d \times d$ matrix $B$, a $d$-dimensional Brownian motion $W$, a Cox process $M^\alpha$ with intensity $\lambda_t^\alpha = \Lambda_1 + \Lambda_2\|\alpha_t\|_2^2$ for constants $\Lambda_1, \Lambda_2 \ge 0$ and independent i.i.d. zero-mean square integrable $d$-dimensional random vectors $Z_j$ with $\mathbb{E}[\|Z_j\|_2^2] = v$, $j = 1, 2, \dots$

Problem (18) is of the general form (1) if instead of the infimum, one considers the supremum of the negative of the expectation. It can be reduced to a one-dimensional ODE which has a closed form solution in the special case $\Lambda_2 = 0$ and admits a very precise numerical solution via a standard ODE solver, such as Runge–Kutta, if $\Lambda_2 > 0$ (see Appendix C). This provides highly accurate reference solutions $V$ and $\alpha^*$ for the value function and optimal control.

Figure 1 compares the performances of GPI-PINN and GPI-CBU on a 10-dimensional ($d = 10$) LQR problem of the form (18) with $T = 1$, $B = I_d$, $c_1 = 1$, $c_2 = 1/4$ without jumps ($\Lambda_1 = \Lambda_2 = 0$) and with jumps ($\Lambda_1 = 0$, $\Lambda_2 = 2$ and $d$-dimensional standard normal jumps $Z_j$). It shows mean absolute errors of $V_{\theta^{(k)}}$ with respect to $V$ given by $\mathrm{MAE}_V = \frac{1}{M}\sum_{i=1}^M |V_{\theta^{(k)}}(t_i, x_i) - V(t_i, x_i)|$ on a test set of size $M$ uniformly sampled from $[0, 1] \times [-2.5, 2.5]^d$ together with runtimes as functions of the number of epochs $k$. It can be seen that GPI-PINN and GPI-CBU exhibit similar convergence in the number of epochs. The residual-based approach of GPI-PINN makes it more stable than GPI-CBU; see also Baird (1995). On the other hand, GPI-CBU has a lower computational cost. This is already the case without jumps (left plot of Figure 1) since it avoids the numerical evaluation of third-order derivatives but becomes much more significant with jumps (right plot of Figure 1) as it also circumvents the numerical integration of the jumps.

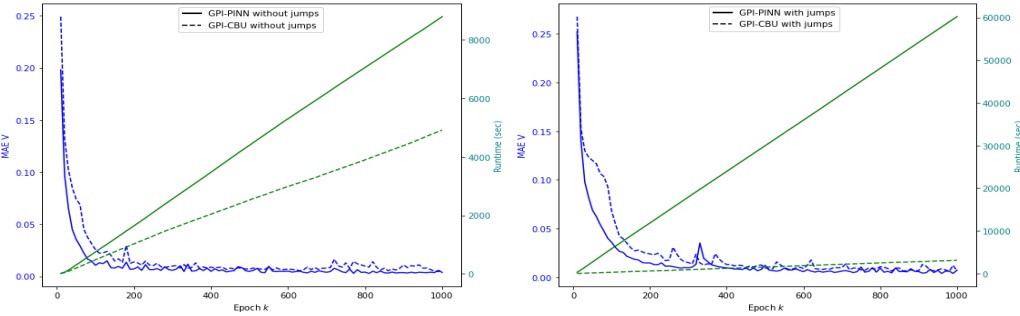

Figure 1: Comparison of $\mathrm{MAE}_V$ (blue) and runtime in seconds (green) of GPI-PINN (solid lines) and GPI-CBU (dashed lines) for a 10-dimensional LQR problem (18) without jumps (left) and with jumps (right).

Figure 2 shows results of GPI-CBU for the same LQR problem with jumps in $d = 50$ dimensions. While the high dimensionality renders GPI-PINN infeasible, GPI-CBU achieves high accuracy in the approximation of the value function as well as the optimal policy. Additional results for up to 150-dimensional LQR problems are reported in Appendix C.

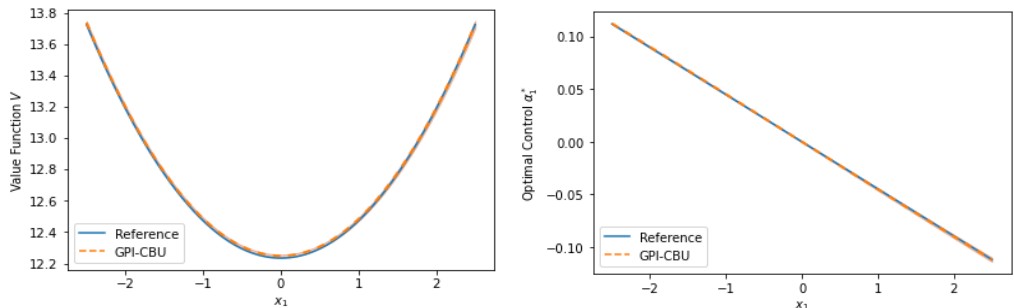

Figure 2: Value function $V(0, x)$ (left) and first component of the optimal control $\alpha_1^*(0, x)$ (right) for $x = (x_1, 0, \ldots, 0)$ with $x_1 \in [-2.5, 2.5]$ for a 50-dimensional LQR problem with jumps. Orange dotted lines: numerical results of GPI-CBU with $\pm 1$ standard deviation given by orange shaded area. Blue lines: reference solutions $V$ and $\alpha^*$.

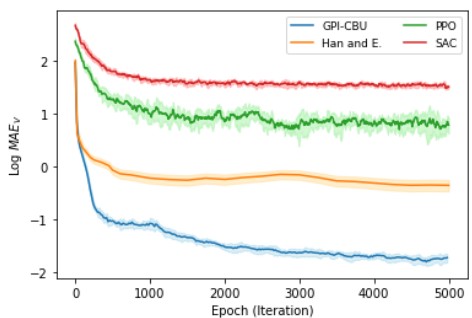

Figure 3: $\log \mathrm{MAE}_V$ of different deep-learning methods for a 10-dimensional LQR problem with jumps.

Figure 3 shows the accuracy of GPI-CBU for the 10-dimensional LQR problem with jumps from the right panel of Figure 1 compared with the two popular model-free RL algorithms PPO and SAC as well as the model-based discrete-time approach of Han & E (2016) applied to a time-discretization of the state dynamics (19). It can be seen that in this setup, PPO and SAC cannot compete with the two model-based approaches since they do not explicitly use the dynamics (19) but instead only rely on sampling from the environment. The method of Han & E (2016) outperforms PPO and SAC but does not achieve the same accuracy as GPI-CBU due to discretization errors and since it does not generalize well to unseen points in the test set. Trying to cover the space-time domain well, we ran the method of Han & E (2016) from several randomly sampled starting points $x_0 \in D$. But being a local method, it tends to learn the optimal control only along the optimal state trajectories $(t, X_t^{\alpha^*})_{0 \le t \le T}$, which results in poor performance in parts of the space-time domain that are not explored well. Additional results are discussed in Appendix C.

## 5.2 Optimal consumption-investment with jumps

As a second example, we consider an economic agent who consumes at relative rate $c_t$ and invests in $n$ financial assets according to a strategy[7] $(\pi_t^1, \ldots, \pi_t^n)$ so as to maximize

$$\mathbb{E}\left[\int_0^T e^{-\rho s} u(c_s Y_s^\alpha) ds + e^{-\rho T} U(Y_T^\alpha)\right] \tag{20}$$

for two utility functions[8] $u, U \colon \mathbb{R}_+ \to \mathbb{R}$, where $Y_t^\alpha$ is the wealth process evolving like

$$\frac{dY_t^\alpha}{Y_{t-}^\alpha} = \left(r_t + \sum_{i=1}^n \pi_t^i (\mu^i - r_t) - c_t\right) dt + \sum_{i=1}^n \pi_t^i \left[\sigma_t^i dW_t^i + d\sum_{j=1}^{M_t^i} \left(e^{Z_j^i} - 1\right)\right] \tag{21}$$

for a stochastic interest rate $r_t$, expected return rates $\mu^i$, stochastic volatilities $\sigma_t^i$, an $n$-dimensional Brownian motion $W$ with correlation matrix $\Sigma$, Cox processes $M^i$ with stochastic jump intensities $\lambda_t^i$ and random jump variables $Z_j^i$. We consider strategies of the form and $c(t, \sigma_t, \lambda_t, r_t, Y_{t-}^\alpha)$ and $\pi^i(t, \sigma_t, \lambda_t, r_t, Y_{t-}^\alpha)$. The problem has $d = 2n + 2$ state variables, consisting of $\sigma_t^i, \lambda_t^i, r_t, Y_t^\alpha$ and

---

[7]$\pi_t^i$ describes the fraction of the agent's total wealth held in the $i^{th}$ asset at time $t$.

[8]In our numerical experiments, we choose CRRA utility functions.

$n+1$ decision variables $c_t, \pi_t^i, i = 1, \ldots, n$. In general, the corresponding HJB equation, given in (35) in Appendix D.2, does not have an analytical solution. But it can be seen in Figure 4 that GPI-CBU converges numerically for $n = 25$ financial assets. More details about this consumption-investment problem with stochastic coefficients are provided in Appendix D.2.

In Appendix D.1, we consider a simplified version of the problem where the coefficients $\sigma_t^i, \lambda_t^i, r_t$ are constant. This case can be reduced to a one-dimensional ODE that can be solved with a standard Runge–Kutta scheme to obtain reference solutions. GPI-CBU yields numerical solutions that are virtually indistinguishable from the Runge–Kutta results.

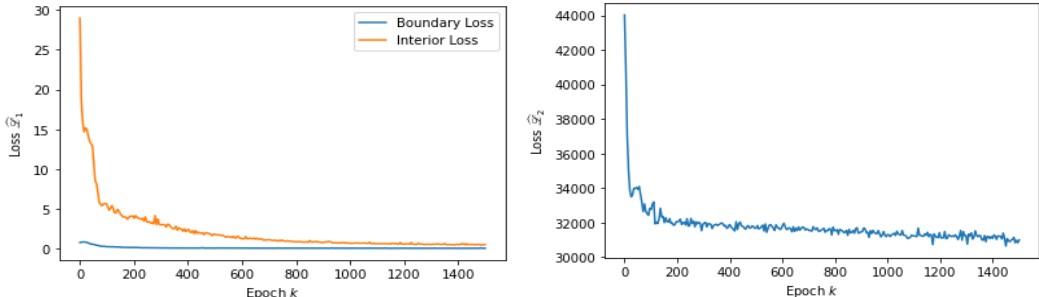

Figure 4: Training losses $\widehat{\mathscr{L}}_1^{(k)}(\theta^{(k+1)})$ (left) and $\widehat{\mathscr{L}}_2^{(k+1)}(\phi^{(k+1)})$ (right) of GPI-CBU as functions of the epoch $k$ for a consumption-investment problem of the form (20)–(21) with $n = 25$ financial assets. The blue curve in the left plot represents the interior loss of $\widehat{\mathscr{L}}_1^{(k)}$ and the orange curve its boundary part; see (14).

## 6 Conclusions, limitations and future work

In this paper, we have introduced two iterative deep learning algorithms for solving finite-horizon stochastic control problems with jumps. Both use an actor-critic approach and train two neural networks to approximate the value function and optimal control, providing global solutions over the entire space-time domain without requiring simulation or discretization of the underlying state dynamics. Our first algorithm, GPI-PINN, works well for high-dimensional problems without jumps but becomes computationally infeasible in the presence of jumps. The second algorithm, GPI-CBU, leverages an efficient expectation-free updating rule based on Proposition 4.1 which makes it particularly well-suited for high-dimensional problems with jumps. Both algorithms are model-based. As such, they outperform model-free RL methods in cases where the underlying state dynamics are known. The accuracy and scalability of the two algorithms has been demonstrated in different numerical examples.

A limitation of our approach lies in the need to know the underlying dynamics of the state process, which are not always available in real-world applications. While it is reasonable to assume that physical systems obey known laws of motion, dynamics in economics and finance typically need to be inferred from data. However, in such cases, they can be learned in a preliminary step using e.g. recent model-learning algorithms such as Brunton et al. (2016) or Champion et al. (2019). Once an approximate model has been learned from data, one of the proposed algorithms, GPI-PINN or GPI-CBU, can be applied to efficiently solve the resulting stochastic control problem.

## Acknowledgments

Financial support from Swiss National Science Foundation Grant 10003723 is gratefully acknowledged. We thank the reviewers for their helpful comments and constructive suggestions.

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

## A Proofs

### A.1 Proof of Proposition 3.1

Denoting $v_i(\cdot) = v\left(t + \frac{h^2}{2n}, x + \frac{h}{\sqrt{2}}\sigma_i(t, x, a) + \frac{h^2}{2n}\beta(t, x, a)\right)$, the second-order derivative of $\psi(h) := \sum_{i=1}^{n} v_i(\cdot)$ is given by

$$\psi''(h) = \sum_{i=1}^{n} \left( \frac{1}{n}\partial_t v_i(\cdot) + \frac{h^2}{n^2}\partial_{tt}^2 v_i(\cdot) + 2\frac{h}{n}\left(\frac{\sigma_i(t, x, a)}{\sqrt{2}} + \frac{h}{n}\beta(t, x, a)\right)^\top \nabla_{tx} v_i(\cdot) \right.$$
$$\left. + \frac{1}{n}\beta^\top(t, x, a)\nabla_x v_i(t, x) + \left(\frac{\sigma_i(t, x, a)}{\sqrt{2}} + \frac{h}{n}\beta(t, x, a)\right)^\top \nabla_x^2 v_i(\cdot) \left(\frac{\sigma_i(t, x, a)}{\sqrt{2}} + \frac{h}{n}\beta(t, x, a)\right) \right).$$

Evaluating it at $h = 0$ gives

$$\psi''(0) = \partial_t v(t, x) + \beta^\top(t, x, a)\nabla_x v(t, x) + \frac{1}{2}\text{Tr}\left[\sigma\sigma^\top(t, x, a)\nabla_x^2 v(t, x)\right],$$

which proves the proposition. □

### A.2 Proof of Proposition 4.1

Since $Y_t$ is independent of $Z_1$, one obtains from the definition of $G_\zeta$ that

$$\mathbb{E}[G_\zeta(t, Y_t, Z_1, V^\alpha, \alpha(t, Y_t)) \mid Y_t = x] = V^\alpha(t, x) + \zeta\Big(\partial_t V^\alpha(t, x) + f(t, x, \alpha(t, x))$$
$$+ \beta^\top(t, x, \alpha(t, x))\nabla_x V^\alpha(t, x) + \frac{1}{2}\text{Tr}\left[\sigma\sigma^\top(t, x, \alpha(t, x))\nabla_x^2 V^\alpha(t, x)\right]$$
$$+ \lambda(t, x, \alpha(t, x))\mathbb{E}\left[V^\alpha(t, x + \gamma(t, x, Z_1, \alpha(t, x))) - V^\alpha(t, x)\right]\Big)$$
$$= V^\alpha(t, x) + \zeta\mathcal{H}(t, x, V^\alpha, \alpha(t, x)) \tag{22}$$
$$= V^\alpha(t, x),$$

where the last equality follows from Theorem 2.1. On the other hand, it is well known that

$$\mathbb{E}[G_\zeta(t, Y_t, Z_1, V^\alpha, \alpha(t, Y_t)) \mid Y_t] = g(Y_t)$$

for the Borel measurable function $g \colon D \to \mathbb{R}$ minimizing the mean squared error

$$\mathbb{E}\left[\left(g(Y_t) - G_\zeta(t, Y_t, Z_1, V^\alpha, \alpha(t, Y_t))\right)^2\right];$$

see e.g. Theorem 4.1.15 of Durrett (2019). This completes the proof. □

## B DGM Architecture

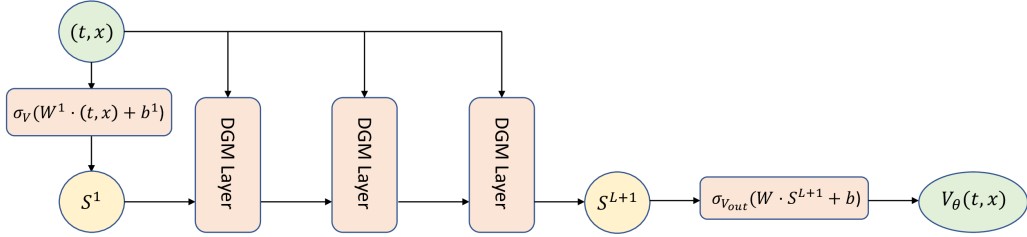

Figure 5: DGM architecture of the value neural network with $L = 3$ (*i.e.* 4 hidden layers).

Each DGM layer in Figure 5 in the value network $V_\theta$ is of the following form

$$
\begin{aligned}
S^1 &= \sigma_V\left(W^1 \cdot (t,x) + b^1\right), \\
Z^\ell &= \sigma_V\left(U^{z,\ell} \cdot (t,x) + W^{z,\ell} \cdot S^\ell + b^{z,\ell}\right), \quad \ell = 1, \ldots, L, \\
G^\ell &= \sigma_V\left(U^{g,\ell} \cdot (t,x) + W^{g,\ell} \cdot S^1 + b^{g,\ell}\right), \quad \ell = 1, \ldots, L, \\
R^\ell &= \sigma_V\left(U^{r,\ell} \cdot (t,x) + W^{r,\ell} \cdot S^\ell + b^{r,\ell}\right), \quad \ell = 1, \ldots, L, \\
H^\ell &= \sigma_V\left(U^{h,\ell} \cdot (t,x) + W^{h,\ell} \cdot \left(S^\ell \odot R^\ell\right) + b^{h,\ell}\right), \quad \ell = 1, \ldots, L, \\
S^{\ell+1} &= \left(1 - G^\ell\right) \odot H^\ell + Z^\ell \odot S^\ell, \quad \ell = 1, \ldots, L, \\
V_\theta(t,x) &= \sigma_{V_{out}}\left(W \cdot S^{L+1} + b\right),
\end{aligned}
$$

where the number of hidden layers is $L+1$, $\cdot$ denotes matrix multiplication and $\odot$ element-wise multiplication. The DGM parameters of the value network are

$$
\theta = \Big\{ W^1, b^1, \left(U^{z,\ell}, W^{z,\ell}, b^{z,\ell}\right)_{\ell=1}^L, \left(U^{g,\ell}, W^{g,\ell}, b^{g,\ell}\right)_{\ell=1}^L,
$$
$$
\left(U^{r,\ell}, W^{r,\ell}, b^{r,\ell}\right)_{\ell=1}^L, \left(U^{h,\ell}, W^{h,\ell}, b^{h,\ell}\right)_{\ell=1}^L, W, b \Big\}.
$$

The number of units in each layer is $N$. $\sigma_V : \mathbb{R}^N \to \mathbb{R}^N$ is a twice-differentiable element-wise nonlinearity and $\sigma_{V_{out}} : \mathbb{R}^N \to \mathbb{V}$ is the output activation function, also twice-differentiable, where $\mathbb{V} \subseteq \mathbb{R}$ is chosen so as to satisfy possible restrictions on the value function's output, resulting from the form of the stochastic control problem (e.g. non-negative value function). The control network is designed following the same DGM architecture with $\sigma_{\alpha_{out}} : \mathbb{R}^N \to A$. Throughout the numerical examples of the paper, we use $L = 3$ with $N = 50$ neurons in each of the DGM layers, see Figure 5. In the value network, we use $\tanh$ for the activation function $\sigma_V$ and softplus for $\sigma_{V_{out}}$. For the control network, we adopt $\tanh$ for $\sigma_\alpha$, while as output activation $\sigma_{\alpha_{out}}$ we choose the identity in Example 5.1 and softplus in Example 5.2.

Unless stated otherwise, we use a number of sample points $M_1, M_2$ equal to 256, a number of gradient steps $N_1, N_2$ equal to 64 and a maximum number of epochs $k_* = 1500$. The network parameters are updated using Adam (Kingma & Ba, 2014) with constant learning rates $\eta_1, \eta_2$ equal to 0.001. Both GPI-PINN and GPI-CBU are fairly robust with respect to these choices. In our setting, the most critical hyperparameters for convergence are the proportionality factors $\xi_1, \xi_2$, which we determined using the approach of Wang et al. (2022), together with the scaling rate $\zeta$ (specific to GPI-CBU). In our experiments, we set $\zeta = 1$ as it provides a good trade-off between convergence speed and accuracy of the improvements in GPI-CBU. Alternatively, one could consider an adaptive scaling factor $\zeta_k > 0$ depending on the epoch $k$. On the other hand, negative scaling factors resulted in non-convergence in all our experiments with exploding losses $\widehat{\mathscr{L}}_1^{(k)}$ and $\widehat{\mathscr{L}}_2^{(k)}$ after only a few epochs.

Algorithms 1 and 2 were implemented using TensorFlow and Keras with GPU acceleration on an NVIDIA RTX 4090.

## C  Linear-quadratic regulator with jumps: detailed results

The value function of the LQR problem with jumps (18) is given by

$$
V(t,x) = \inf_\alpha \mathbb{E}\left[\int_0^T c_1 \|\alpha_s\|_2^2 \, ds + c_2 \|X_T^\alpha\|_2^2 \;\Big|\; X_t^\alpha = x\right]. \tag{23}
$$

The corresponding HJB equation is

$$
\begin{aligned}
0 = \partial_t V(t,x) + \frac{1}{2}\mathrm{Tr}\left[BB^\top \nabla^2 V_x(t,x)\right] + \inf_{a \in \mathbb{R}^d} \Big\{ &c_1 \|a\|_2^2 + a^\top \nabla_x V(t,x) \\
&+ \left(\Lambda_1 + \Lambda_2 \|a\|_2^2\right) \mathbb{E}\left[V(t, x + Z_1) - V(t,x)\right]\Big\}
\end{aligned}
$$

with terminal condition $V(T,x) = c_2 \|x\|_2^2$. Using the Ansatz

$$
V(t,x) = g(t) + \frac{1}{2} h(t) \|x\|_2^2,
$$

it is straightforward to see that $g$ is of the form

$$g(t) = \frac{1}{2}\big(\text{Tr}[BB^\top] + \Lambda_1 v\big) \int_t^T h(s)ds \tag{24}$$

for the unique solution of the non-linear ODE

$$h'(t) = \frac{h^2(t)}{2c_1 + h(t)v\Lambda_2}, \quad h(T) = 2c_2. \tag{25}$$

(25) can efficiently be solved using a numerical method such as Runge–Kutta of order 5(4); see e.g. Dormand & Prince (1980). The optimal control $\alpha^*$ is given in terms of $h$ by

$$\alpha^*(t,x) = -\frac{h(t)x}{2c_1 + h(t)v\Lambda_2}.$$

Note that in the constant intensity case $\Lambda_1 > 0$, $\Lambda_2 = 0$, (24)–(25) admit the explicit solutions

$$g(t) = \big(\text{Tr}[BB^T] + \Lambda_1 v\big)c_1\left(\log(\frac{c_2}{c_1}(T-t)+1)\right) \quad \text{and} \quad h(t) = \frac{2c_1c_2}{c_1 + c_2(T-t)}. \tag{26}$$

But also in the case $\Lambda_2 > 0$, numerical solutions of (24)–(25) obtained with Runge–Kutta of order 5(4) provide very precise reference solutions $V$ and $\alpha^*$ to the LQR problem (18).

GPI-PINN and GPI-CBU both need random points $(t_m, x_m) \in [0,T] \times \mathbb{R}^d$ and $(y_m) \in \mathbb{R}^d$ sampled from two distributions $\mu$ on $[0,T] \times \mathbb{R}^d$ and $\nu$ on $\mathbb{R}^d$, which we chose according to Bachouch et al. (2022).

In the following we consider problem (18) with $T = 1$, $B = I_d$, $c_1 = 1$, $c_2 = 1/4$ and $d$-dimensional standard normal jumps $Z_j$. Figures 6–8 show results obtained with GPI-CBU in $d = 50$ dimensions for the constant jump intensity case $\Lambda_1 = 1/4$ and $\Lambda_2 = 0$, which admits analytical reference solutions.

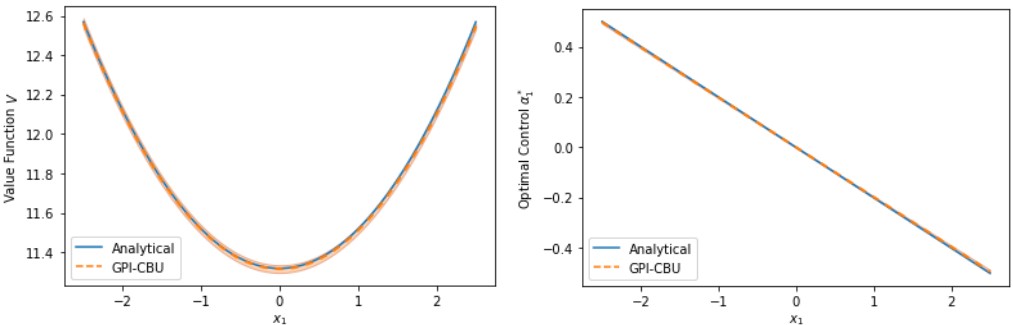

Figure 6: Value function $V(0,x)$ (left) and first component of the optimal control $\alpha_1^*(0,x)$ (right) for $x = (x_1, 0, \ldots, 0)$ with $x_1 \in [-2.5, 2.5]$. Orange dotted lines: numerical results of GPI-CBU with $\pm 1$ standard deviation given by orange shaded area. Blue lines: analytical solution (26). $d = 10$, $\Lambda = 1/4$, $\Lambda_2 = 0$, $k_* = 1500$.

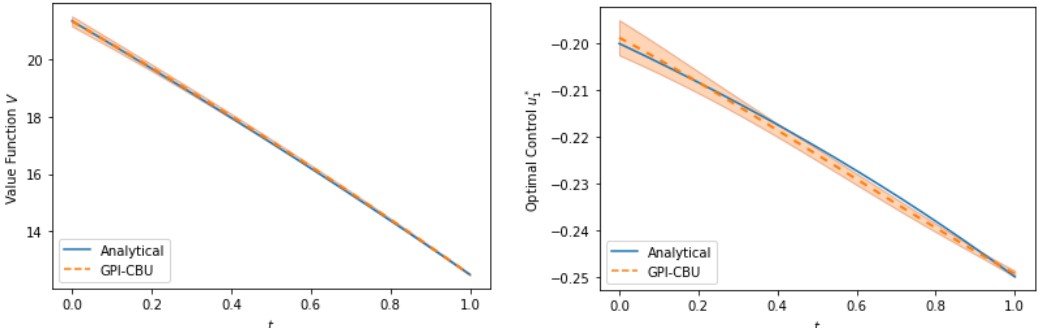

Figure 7: Value function $V(t,x)$ (left) and first component of the optimal control $\alpha_1^*(t,x)$ (right) for $t \in [0,1]$ and $x = \mathbf{1}_{50}$. Orange dotted lines: numerical results from GPI-CBU with $\pm 1$ standard deviation in orange shaded area. Blue lines: analytical solution (26). $d = 50$, $\Lambda_1 = 1/4$, $\Lambda_2 = 0$, $k_* = 1500$.

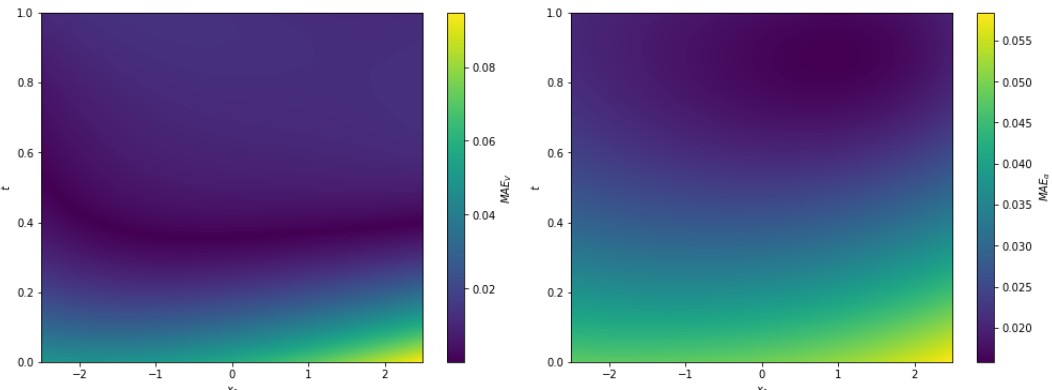

Figure 8: Heatmaps of $\text{MAE}_V$ (left) and $\text{MAE}_\alpha$ (right) for $t \in [0,1]$ and $x = (x_1, 1, \ldots, 1)$ with $x_1 \in [-2.5, 2.5]$ obtained from GPI-CBU for the LQR problem with $d = 50$, $\Lambda_1 = 1/4$ and $\Lambda_2 = 0$. $k_* = 1500$.

$\text{MAE}_\alpha$ in Figure 8 is defined, analogously to $\text{MAE}_V$ in Section 5.1, by

$$\text{MAE}_\alpha = \frac{1}{M} \sum_{i=1}^{M} \|\alpha_{\phi^{(k_*)}}(t_i, x_i) - \alpha^*(t_i, x_i)\|_2$$

for a test set of size $M$ uniformly sampled from $[0,1] \times [-2.5, 2.5]^d$.

Next, we consider the same LQR problem (18) as above in 50 dimensions with a controlled jump intensity. Figures 9–10 (and 2), show results of GPI-CBU for $\Lambda_1 = 0$ and $\Lambda_2 = 2$. Reference solutions were computed with Runge–Kutta of order 5(4).

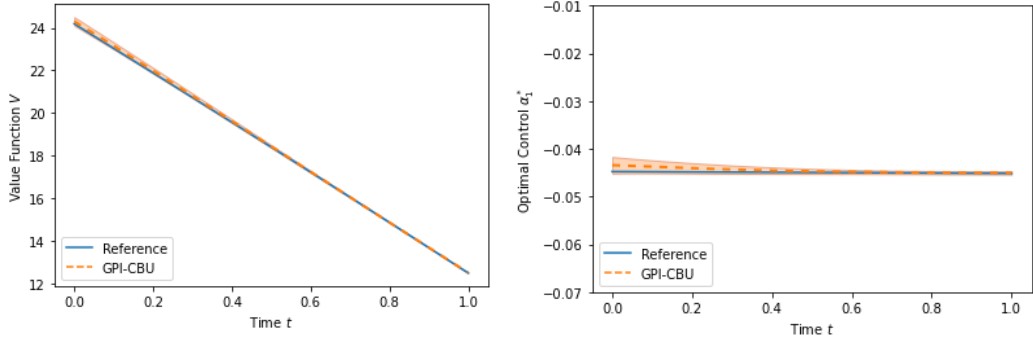

Figure 9: Value function $V(t, x)$ (left) and first component of the optimal control $\alpha_1^*(t, x)$ (right) for $t \in [0,1]$ and $x = \mathbf{1}_{50}$. Orange dotted lines: numerical results of GPI-CBU with $\pm 1$ standard deviation in orange shaded area. Blue lines: reference solution. $d = 50$, $\Lambda_1 = 0$, $\Lambda_2 = 2$, $k_* = 1500$.

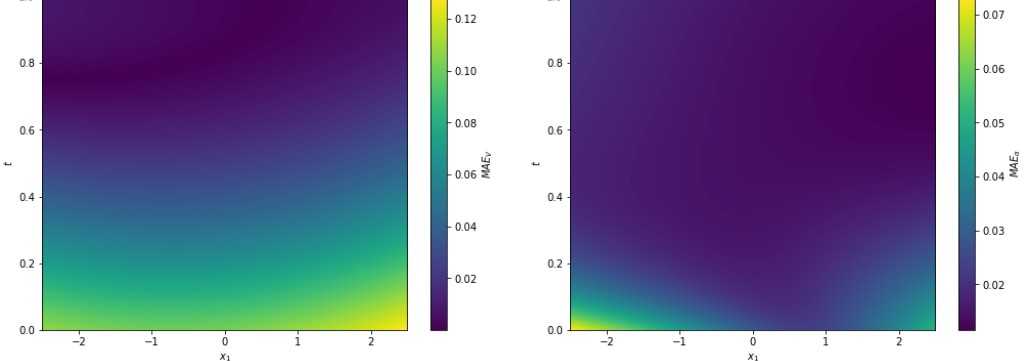

Figure 10: Heatmaps of $\text{MAE}_V$ (left) and $\text{MAE}_\alpha$ (right) for $t \in [0,1]$ and $x = (x_1, \mathbf{1}_{49})$ with $x_1 \in [-2.5, 2.5]$ obtained from GPI-CBU for the LQR problem with $d = 50$, $\Lambda_1 = 1/4$ and $\Lambda_2 = 0$. $k_* = 1500$.

These results illustrate the accuracy of GPI-CBU for high-dimensional control problems with controlled jumps. It can be seen that the optimal control $\alpha^*$ is lower in magnitude compared to the case of constant jump intensity since now, exerting control increases the likelihood of jumps. But in both cases, it is optimal to steer the state process $X_t^\alpha$ towards 0 as $t$ approaches the time horizon $T$.

Table 1 summarizes results of GPI-CBU for higher-dimensional LQR problems with the same parameters as above with controlled jump intensity $\Lambda_1 = 0$, $\Lambda_2 = 2$ obtained over $k_* = 5000$ training epochs. This demonstrates the scalability of GPI-CBU.

Table 1: GPI-CBU performance metrics as a function of the state dimension $d$ for LQR problems of the form (18) with $\Lambda_1 = 0$, $\Lambda_2 = 2$ and $k_* = 5000$ training epochs.

| Dimensions $d$ | $\mathrm{MAE}_V$ | $\mathrm{MAE}_\alpha$ | Loss $\widehat{\mathscr{L}_1}(\theta^{k_*})$ | Loss $\widehat{\mathscr{L}_2}(\theta^{k_*})$ | Time (sec) |
|---|---|---|---|---|---|
| 5 | 0.0023 | 0.0041 | 0.0237 | -0.1332 | 6,410 |
| 10 | 0.0025 | 0.0049 | 0.0314 | -0.1972 | 8,093 |
| 50 | 0.0147 | 0.0075 | 0.1267 | -0.4206 | 16,129 |
| 100 | 0.0492 | 0.00539 | 0.6970 | -0.4461 | 24,359 |
| 150 | 0.0979 | 0.0096 | 3.7210 | -0.4671 | 33,120 |

# D  Optimal consumption-investment problem

## D.1  Constant coefficients

We first study the optimal consumption-investment problem in a model with constant coefficients, where the risk-free asset evolves according to

$$dS_t^0 = rS_t^0 dt \,, \tag{27}$$

for a constant interest rate $r \in \mathbb{R}$, and there are $n$ stocks with prices following

$$\frac{dS_t^i}{S_{t-}^i} = \mu^i dt + \sigma^i dW_t^i + d\sum_{j=1}^{M_t^i} \left( e^{Z_j^i} - 1 \right) , \tag{28}$$

for constant drifts $\mu^i \in \mathbb{R}$ and volatilities $\sigma^i \in \mathbb{R}_+$, an $n$-dimensional Brownian motion $W$ with correlation matrix $\Sigma$, independent Poisson processes $M^i$ with constant intensities $\lambda^i \geq 0$ and i.i.d. normal random variables $Z_1^i, Z_2^i, \ldots \sim \mathcal{N}(\mu_Z^i, \sigma_Z^i)$ with $\mu_Z^i \in \mathbb{R}$ and $\sigma_Z^i \in \mathbb{R}_+$. Suppose an investor starts with an initial endowment of $Y_0 > 0$, consumes at rate $c_t Y_{t-}^\alpha$ and invests $\pi_t^i Y_{t-}^\alpha$, $i = 1, \ldots n$, in the $n$ stocks. If the risk-free asset is used to balance the transactions, the resulting wealth evolves according to

$$\frac{dY_t^\alpha}{Y_{t-}^\alpha} = \left( r + \sum_{i=1}^n \pi_t^i(\mu^i - r) - c_t \right) dt + \sum_{i=1}^n \pi_t^i \left[ \sigma^i dW_t^i + d\sum_{j=1}^{M_t^i} \left( e^{Z_j^i} - 1 \right) \right] . \tag{29}$$

Let us assume the investor attempts to maximize

$$\mathbb{E}\left[ \int_0^T e^{-\rho s} u(c_s Y_s^\alpha) ds + e^{-\rho T} U(Y_T^\alpha) \right]$$

for two utility functions $u, U \colon \mathbb{R}_+ \to \mathbb{R}$ and a discount factor $\rho \in \mathbb{R}$. Since the driving noise in (29) has stationary and independent increments, it is enough to consider strategies of the form $c_t = c(t, Y_{t-}^\alpha)$ and $\pi_t^i = \pi^i(t, Y_{t-}^\alpha)$ for functions $c, \pi^i \colon [0, T) \times \mathbb{R}_+ \to \mathbb{R}_+$, $i = 1, \ldots, n$. We write this $n + 1$-dimensional control as $\alpha_t = (c_t, \pi_t^1, \ldots, \pi_t^n)^\top \in A := \mathbb{R}_+^{n+1}$. Assuming constant relative risk aversion (CRRA) utility functions $u, U$ with relative risk aversion $\gamma > 0$ and denoting $\delta = 1 - \gamma$, the resulting reward functional is

$$V^\alpha(t, y) = \mathbb{E}\left[ \int_t^T e^{-\rho(s-t)} \frac{(c_s Y_s^\alpha)^\delta}{\delta} ds + e^{-\rho(T-t)} \frac{(Y_T^\alpha)^\delta}{\delta} \,\Big|\, Y_t^\alpha = y \right] . \tag{30}$$

Finally, writing $\mu = (\mu^1, \ldots, \mu^n)^\top$, $\sigma = \mathrm{diag}(\sigma^1, \ldots, \sigma^n)$, $\mu_Z = (\mu_Z^1, \ldots, \mu_Z^n)^\top$, $\sigma_Z = (\sigma_Z^1, \ldots, \sigma_Z^n)^\top$, $\lambda = (\lambda^1, \ldots, \lambda^n)^\top$, $\pi = (\pi^1, \ldots, \pi^n)^\top$, the associated HJB equation for the value function $V(t, y) = \sup_\alpha V^\alpha(t, y)$ satisfies[9] for all $(t, y) \in [0, T) \times \mathbb{R}_+$,

$$
\begin{aligned}
0 = \partial_t V(t, y) - \rho V(t, y) + \sup_{(c, \pi) \in A} \Bigg\{ & \left( r + (\mu - r)^\top \pi - c \right) y\, \partial_y V(t, y) \\
& + \frac{1}{2} \pi^\top \sigma \Sigma \sigma^\top \pi\, y^2\, \partial_{yy}^2 V(t, y) + \sum_{i=1}^n \lambda^i \mathbb{E}\left[ V(t, y + y\,\pi^i(e^{Z_1^i} - 1)) - V(t, y) \right] + \frac{(cy)^\delta}{\delta} \Bigg\},
\end{aligned}
\tag{31}
$$

with $V(T, y) = y^\delta / \delta$. This stochastic control problem does not admit an analytical solution but we can instead characterize its solution in terms of a PIDE, so as to assess the accuracy of the proposed Algorithms 1 and 2. Following Øksendal & Sulem (2007), we assume the value function is of the form $V(t, y) = A(t) y^\delta / \delta$. Then optimizing the HJB equation (31) leads to first order conditions showing that the optimal consumption rate $c^*$ is independent from the wealth $y$ and given by

$$
c^*(t, y) = A(t)^{1/(\delta - 1)},
\tag{32}
$$

and the optimal investment strategy is given by a vector of constants $\pi^*(t, y) = \pi^* := (\pi^{*,1}, \ldots, \pi^{*,n})^\top$ satisfying

$$
(\mu - r)^\top + \sigma \Sigma \sigma^\top \pi^* (\delta - 1) + \sum_{i=1}^n \lambda^i \mathbb{E}\left[ \left( 1 + \pi^{*,i}(e^{Z_1^i} - 1) \right)^{\delta - 1} \left( e^{Z_1^i} - 1 \right) \right] = 0.
$$

Plugging these optimal controls back into the HJB equation (31), we find that the function $A(t)$ satisfies the ODE

$$
\begin{aligned}
\frac{\partial_t A(t)}{A(t)} = \rho - \delta \left( r + (\mu - r)^\top \pi^* - A(t)^{1/(\delta - 1)} \right) - \frac{1}{2} \delta(\delta - 1) \pi^{*\top} \sigma \Sigma \sigma^\top \pi^* \\
- \sum_{i=1}^n \lambda^i \mathbb{E}\left[ \left( 1 + \pi^{*,i}(e^{Z_1^i} - 1) \right)^\delta - 1 \right] - A(t)^{1/(\delta - 1)},
\end{aligned}
\tag{33}
$$

with $A(T) = 1$, which can be solved numerically using again the Runge–Kutta method of order 5(4). For both Algorithms 1 and 2, we sample the time and space points independently from the uniform distributions $\mathcal{U}_{[0,T]}$ and $\mathcal{U}_{[0,y_b]}$, respectively. The parameters of the optimal investment problem below are as follows: $T = 1, y_b = 150, r = 0.02, \rho = 0.045, \delta = 0.7, \lambda = 0.45 \cdot \mathbf{1}_n, \mu_Z = 0.25 \cdot \mathbf{1}_n, \sigma_Z = 0.2 \cdot \mathbf{1}_n, \mu = 0.032 \cdot \mathbf{1}_n, \sigma = I_n, \Sigma = 0.2 \cdot \mathbf{1}_{n \times n}$ with $\mathrm{diag}(\Sigma) = \mathbf{1}_n$. Note that, compared to the LQR problem in Section 5.1, the proportionality factor $\xi$ is now set to 10. Moreover, since $A = \mathbb{R}_+^{n+1}$, $\sigma_{\alpha_{out}}$ is chosen to be the softplus activation function. Instead of the DGM architecture, we here train a classical feedforward neural network with 4 hidden layers, each of 128 neurons. Finally, the $\mathrm{MAE}_V$ and $\mathrm{MAE}_\alpha$ metrics are again defined as in Section 5.1 on a test set of size $M$, uniformly sampled from $[0, 1] \times [0, y_b]$ with $V$ and $\alpha^*$ obtained from the Runge–Kutta method described above.

In Figure 11 we again compare $\mathrm{MAE}_V$ and runtime of GPI-PINN and GPI-CBU as functions of the number of epochs $k$ for $n = 10$ stocks in the consumption-investment problem (30) with and without jumps. We again see that the residual-based GPI-PINN Algorithm 1 tends to be more stable and accurate for larger epochs, despite having a higher runtime. When accounting for jumps in the dynamics (29), GPI-PINN becomes numerically very time-consuming, even for a 10-dimensional control problem. This issue is even amplified compared with the LQR problem (see Figure 1), as the jump size now depends on the control $\pi$ in the wealth dynamics (29). In contrast, GPI-CBU again handles these jumps far more efficiently.

---

[9]Note that the term $-\rho V(t, y)$ in (31) is not contained in our general HJB equation (3). But GPI-PINN and GPI-CBU still work if it is added.

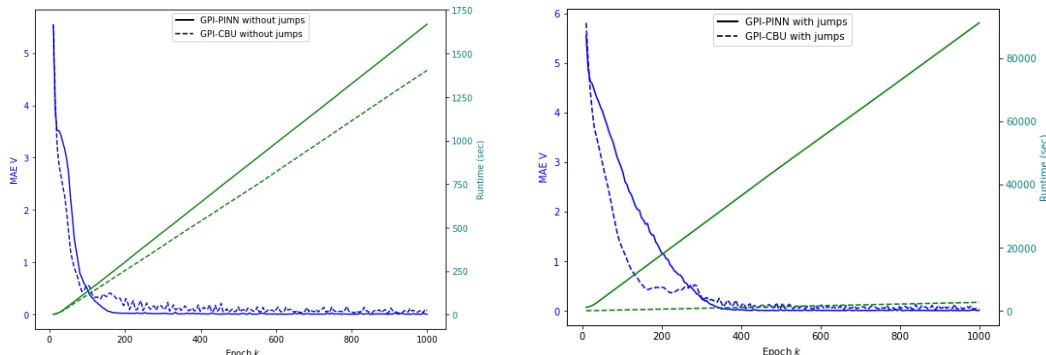

Figure 11: Comparison of $\mathrm{MAE}_V$ (blue) and runtime in seconds (green) of GPI-PINN (solid line) and GPI-CBU (dashed line) as functions of the number of epochs $k$ for the optimal consumption-investment problem without jumps (left) and with jumps (right) for $n = 10$ stocks.

Next, we address a higher-dimensional optimal consumption-investment problem with jumps ($n = 50$ stocks), where only GPI-CBU is implemented since GPI-PINN becomes then numerically infeasible. Figures 12 and 13 again confirm the accuracy of GPI-CBU for both the value function and the optimal consumption rate $c^*(t)$, with the standard deviation across 10 independent runs of GPI-CBU being virtually imperceptible. The optimal wealth allocations $\pi^*$, being constant, are not depicted. However, the $\mathrm{MAE}_\alpha$ heatmap in Figure 14 corroborates our method's accuracy in determining both $c^*$ and $\pi^*$. We also observe that the higher absolute errors tend to occur at the boundaries of the domain.

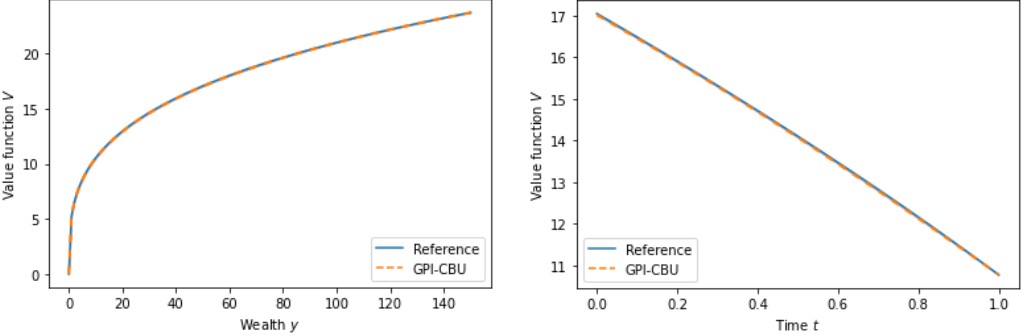

Figure 12: Value function $V(t, y)$ for $t = 0$, $y \in [0, 150]$ (left) and $t \in [0, 1]$, $y = 50$ (right) for the optimal consumption-investment problem (20) with constant coefficients and $n = 50$ stocks. Orange dotted lines: results of GPI-CBU with $\pm 1$ standard deviation given by orange shaded area ($k_* = 1000$). Blue lines: reference solution from Runge–Kutta applied to (32)-(33).

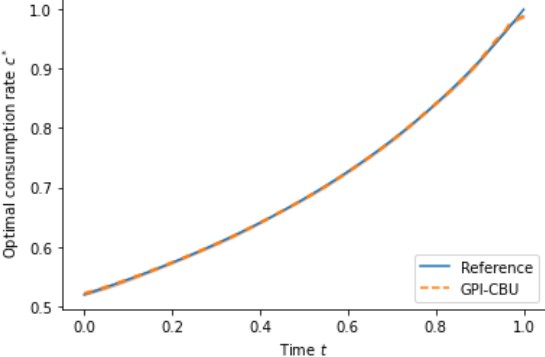

Figure 13: Optimal relative consumption rate $c^*(t)$ for $t \in [0, 1]$ for the optimal consumption-investment problem (20) with constant coefficients and $n = 50$ stocks. Orange dotted line: results of GPI-CBU with $\pm 1$ standard deviation given by orange shaded area ($k_* = 1000$). Blue line: reference solution from Runge–Kutta applied to (32)-(33).

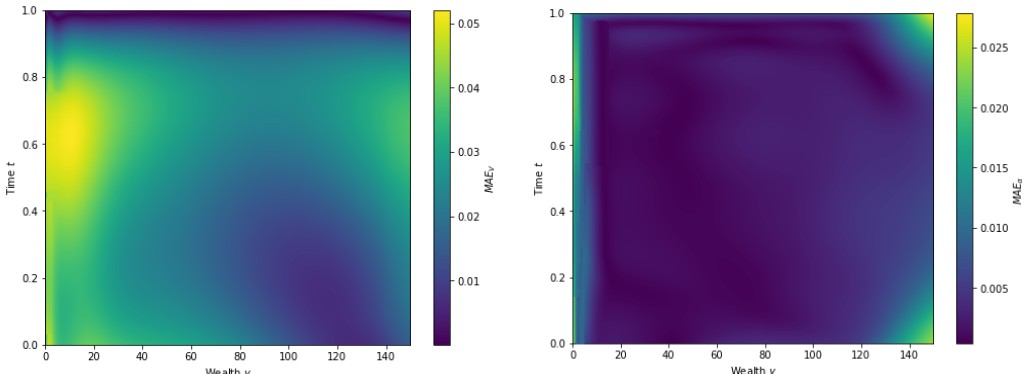

Figure 14: Heatmaps of $\mathrm{MAE}_V$ (left) and $\mathrm{MAE}_\alpha$ (right) for $t \in [0,1]$ and $y \in [0,150]$, for GPI-CBU applied to the optimal consumption-investment problem (20) with constant coefficients and $n = 50$ stocks.

## D.2 Stochastic coefficients

Now, we study a more complex optimal consumption-investment problem in a realistic market of the form (27)–(28) with a stochastic interest rate

$$dr_t = a_r(b_t - r_t)dt + \nu_r\sqrt{r_t}\,dW_t^r,$$

with stochastic (Heston) volatility models for the $n$ stock prices

$$\frac{dS_t^i}{S_{t-}^i} = \mu^i dt + \sigma_t^i\,dW_t^{S,i} + d\sum_{j=1}^{M_t^i}\left(e^{Z_j^i} - 1\right),$$

for $\sigma_t^i = \sqrt{v_t^i}$, where $v_t^i$ follows

$$dv_t^i = a_v^i(b_v^i - v_t^i) + \nu_v^i\sqrt{v_t^i}\,dW_t^{v,i}, \tag{34}$$

and with stochastic jump intensities

$$d\lambda_t^i = a_\lambda^i(b_\lambda^i - \lambda_t^i) + \nu_\lambda^i\sqrt{\lambda_t^i}\,dW_t^{\lambda,i}.$$

We denote $W = (W^{S,1},\ldots,W^{S,n},W^{v,1},\ldots,W^{v,n},W^{\lambda,1},\ldots,W^{\lambda,n},W^r)^\top$ a $(3n+1)$-dimensional Brownian motion with correlation matrix $\Sigma$. In this case, the wealth process can still be described as follows

$$\frac{dY_t^\alpha}{Y_{t-}^\alpha} = \left(r_t + \sum_{i=1}^n \pi_t^i(\mu^i - r_t) - c_t\right)dt + \sum_{i=1}^n \pi_t^i\left[\sqrt{v_t^i}\,dW_t^{S,i} + d\sum_{j=1}^{M_t^i}\left(e^{Z_j^i} - 1\right)\right],$$

and the strategies should be of the form $c(t, v_t, \lambda_t, r_t, Y_{t-}^\alpha)$ and $\pi^i(t, v_t, \lambda_t, r_t, Y_{t-}^\alpha)$, with $\alpha_t = (c_t, \pi_t^1, \ldots, \pi_t^n)^\top \in A := \mathbb{R}_+^{n+1}$. Hence, the dynamics of the $(2n+2)$-dimensional process $X_t^\alpha = (v_t, \lambda_t, r_t, Y_{t-}^\alpha)$ are given by

$$dX_t^\alpha = \mu_X(t, X_t^\alpha)dt + \Sigma_X(t, X_t^\alpha)dW_t + \gamma_X(t, X_t^\alpha)\,d\sum_{i=1}^n\sum_{j=1}^{M_t^i}\left(e^{Z_j^i} - 1\right),$$

with $\mu_X(\cdot) \in \mathbb{R}^{2n+2}$, $\Sigma_X(\cdot) \in \mathbb{R}_+^{(2n+2)\times(3n+1)}$ and $\gamma_X(\cdot) \in \mathbb{R}^{2n+2}$ such that

$$\mu_X(t, X_t^\alpha) = \begin{pmatrix} a_v(b_v - v_t) \\ a_\lambda(b_\lambda - \lambda_t) \\ a_r(b_r - r_t) \\ Y_{t-}^\alpha(r_t + (\mu - r_t)^\top\pi_t - c_t) \end{pmatrix}, \quad \gamma_X(t, X_t^\alpha) = \begin{pmatrix} \mathbf{0}_n \\ \mathbf{0}_n \\ 0 \\ Y_{t-}^\alpha\pi_t^\top\mathbf{1}_n \end{pmatrix},$$

and

$$\Sigma_X(t, X_t^\alpha) = \begin{pmatrix} \mathbf{0}_{n\times n} & D_v & \mathbf{0}_{n\times n} & \mathbf{0}_{n\times 1} \\ \mathbf{0}_{n\times n} & \mathbf{0}_{n\times n} & D_\lambda & \mathbf{0}_{n\times 1} \\ \mathbf{0}_{1\times n} & \mathbf{0}_{1\times n} & \mathbf{0}_{1\times n} & \nu_r\sqrt{r_t} \\ D_Y & \mathbf{0}_{1\times n} & \mathbf{0}_{1\times n} & 0 \end{pmatrix},$$

where $D_v = \mathrm{diag}\left(\nu_v^1\sqrt{v_t^1}, \ldots, \nu_v^n\sqrt{v_t^n}\right)_{n\times n}$, $D_\lambda = \mathrm{diag}\left(\nu_\lambda^1\sqrt{\lambda_t^1}, \ldots, \nu_\lambda^n\sqrt{\lambda_t^n}\right)_{n\times n}$, $D_Y = \left(\pi_t^1 Y_{t-}^\alpha\sqrt{v_t^1}, \ldots, \pi_t^n Y_{t-}^\alpha\sqrt{v_t^n}\right)_{1\times n}$. Therefore, the value function

$$V(t, x) = \sup_\alpha \mathbb{E}\left[\int_t^T e^{-\rho(s-t)}\frac{(c_s Y_s^\alpha)^\delta}{\delta}ds + e^{-\rho(T-t)}\frac{(Y_T^\alpha)^\delta}{\delta}\,\bigg|\, X_t^\alpha = x\right],$$

satisfies the following HJB equation for all $(t, x) \in [0, T) \times \mathbb{R}_+^n \times \mathbb{R}_+^n \times \mathbb{R} \times \mathbb{R}_+$,

$$0 = \partial_t V(t, x) - \rho V(t, x) + \sup_{(c,\pi)\in A}\left\{\mu_X^\top(t, x)\,\nabla_x V(t, x) + \frac{1}{2}\mathrm{Tr}\left[\Sigma_X(t, x)\Sigma\Sigma_X^\top(t, x)\,\nabla_x^2 V(t, x)\right]\right.$$

$$\left. + \sum_{i=1}^n \lambda^i\,\mathbb{E}\left[V(t, v, \lambda, r, y + y\,\pi^i(e^{Z_1^i} - 1)) - V(t, x)\right] + \frac{(cy)^\delta}{\delta}\right\}, \tag{35}$$

with terminal condition $V(T, x) = y^\delta/\delta$. We consider a portfolio of $n = 25$ stocks, resulting in a 52-dimensional value function and a 26-dimensional control process, with the same parameters as in Section D.1. This version of the consumption-investment problem is significantly more complex than the one discussed in Section D.1, with the value function's dimensionality increasing from 2 to 52. Consequently, Runge–Kutta can no longer be used to solve the associated HJB equation (35), making it impossible to compute the $\mathrm{MAE}_V$ and $\mathrm{MAE}_\alpha$ metrics. However, despite this lack of a reference for comparison, GPI-CBU still produces results that appear reasonable. We indeed first observe in Figure 4 (main manuscript) that the losses $\widehat{\mathscr{L}}_1^{(k)}$ (14) and $\widehat{\mathscr{L}}_2^{(k)}$ (16) converge as the number of epoch $k$ increases. Note that the orange curve in the left plot represents the interior loss (first right-hand term) of $\widehat{\mathscr{L}}_1^{(k)}$, while the blue curve is the boundary loss (second right-hand term) of $\widehat{\mathscr{L}}_1^{(k)}$, see Eq. (14).

The following Figures 15 and 16 confirm that the results of GPI-CBU for both the value function and optimal control are reasonable, as they closely resemble those of Figures 12 and 13. For the value functions of Figure 15, the standard deviations across 10 independent runs of GPI-CBU remain very low, confirming the stability of the approximations. For the optimal consumption rate in Figure 16 (left plot), the standard deviation across the 10 runs tends to be higher for values of time $t$ close to 0, although being still reasonable. Finally, varying the first dimension of the intensity $\lambda^1$ mainly impacts the corresponding fraction of wealth $\pi_t^{*,1}$, while its effect on the other proportions and consumption rate is more moderate (since arising from the correlation matrix $\Sigma$ of the Brownian motion $W$).

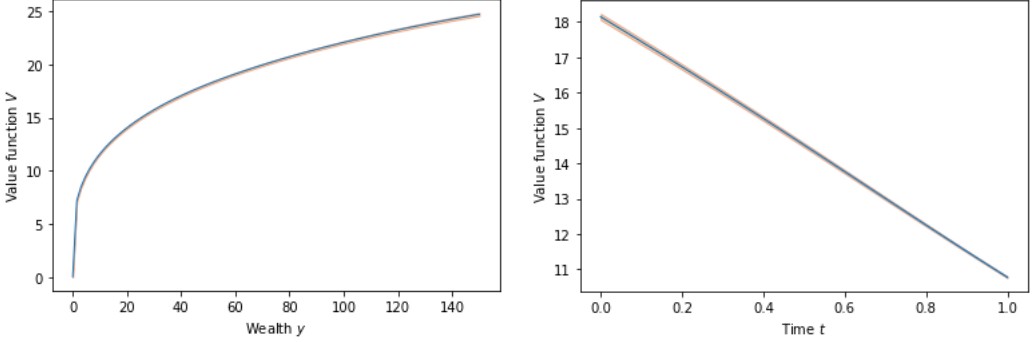

Figure 15: Value function $V(t, x)$ for $t = 0$ and $x = (0.15 \cdot \mathbf{1}_{10}, \mathbf{1}_{10}, 0.02, y)$ with $y \in [0, 150]$ (left) and for $t \in [0, 1]$ and $x = (0.15 \cdot \mathbf{1}_{10}, \mathbf{1}_{10}, 0.02, 50)$ (right), for the optimal consumption-investment problem (20) with stochastic coefficients and $n = 25$ stocks. Blue lines: results of GPI-CBU with $\pm 1$ standard deviation given by orange shaded area ($k_* = 1000$).

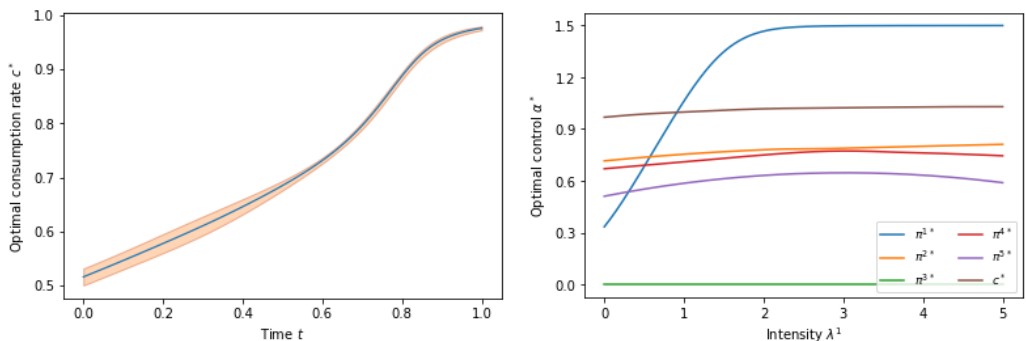

Figure 16: Optimal policy for the optimal consumption-investment problem (20) with stochastic coefficients and $n = 25$ stocks. Left plot: optimal consumption rate $c^*(t, x)$ for $t \in [0, 1]$ and $x = (0.15 \cdot \mathbf{1}_{10}, \mathbf{1}_{10}, 0.02, 50)$, results of GPI-CBU in blue and $\pm 1$ standard deviation given by orange shaded area ($k_* = 1000$). Right plot: optimal consumption rate $c^*(t, x)$ and first five optimal fractions of wealth $\pi^*(t, x)$ for $t = 0$ and $x = (0.15 \cdot \mathbf{1}_{10}, \lambda^1, \mathbf{1}_9, 0.02, 50)$ for $\lambda^1 \in [0, 5]$.

