# OpenReview forum: "Deep learning for continuous-time stochastic control with jumps"
_NeurIPS.cc/2025/Conference — NeurIPS 2025 poster_

### Official Review · Reviewer_9fvu · 2025-06-30

**Clarity:** 2
**Significance:** 3
**Originality:** 3
**Rating:** 4
**Confidence:** 3

**Summary:**

This paper introduces a deep learning approach to solve stochastic optimal control problems with jumps. The method can be interpreted as a PINN-style approach to solving the HJB of the optimal control problem. Since the HJB is second order, a PINN style approach would require computing third-order derivatives, which quickly becomes infeasible. The paper proposes a second algorithm that circumvents this issue using a recursive approach instead of relying solely on HJB residuals. Theoretical results are accompanied with numerical results on small dimension settings.

**Questions:**

1. Can the authors provide a condensed summary of differences between algorithm 1 and 2?
2. The experiments for consumption-investment do not compare the proposed algorithm with other standard DL frameworks as done for the LQR setting. Could the authors share these experimental results as well?
3. Could the authors comment on the scalability and potential of this approach to larger scale environments? Are there environments or settings that are particularly suited to the strengths of the proposed algorithms?

**Ethical Concerns:**

["NO or VERY MINOR ethics concerns only"]

**Final Justification:**

While the authors did share improved results for slightly higher dimensionality, certain concerns from other reviewers regarding theoretical concerns make me unsure of raising my score, hence I shall maintain my score of 4.

**Limitations:**

Yes, the authors have adequately addressed the limitations of their work.

**Quality:**

3

**Strengths And Weaknesses:**

Strengths:
1. Theoretical results are sound and proofs seem correct.
2. The main algorithms provided are original and introduce a new approach to solving optimal control problems.
3. The writing is thorough and comprehensive, and the mathematical results are precise.

Weaknesses:
1. Some of the exposition and notation is extremely dense and hard to follow.
2. Numerical experiments seem to only address dimensions d=10 to 50 and are limited to LQR and consumption-investment problem. It would be nice to mention larger scale problems where this could be applied.
3. The quality of the plots is unclear and makes it hard to read in some cases.

---

> ### Author Rebuttal · Authors · 2025-07-25
>
> **Q1:**  *Can the authors provide a condensed summary of differences between algorithm 1 and 2?*
>
> **Answer:** Sure, both are actor-critic methods using two neural networks $V_\theta$ and $\alpha_\phi$ to approximate the value function and optimal control, respectively.
>
> GPI-PINN 1 uses a PINN approach to approximate the value function by minimizing the residuals $\mathcal{H}$ of the HJB equation (4). It works well in low dimensions and without jumps but becomes inefficient otherwise since it needs to compute the Hessian and the jump-expectations $\mathbb{E}^{\mathcal{Z}} V_\theta(t, x + \gamma(t,x,Z_1,\alpha_\phi(t,x))$ in $\mathcal{H}$ for each sample $(t,x)$ in every iteration. Additionally, it requires the backpropagation through the second-order Hamiltonian $\nabla_\theta \mathcal{H}$, see Eq. (20), and hence to evaluate third-order derivatives, which makes it more computationally intensive.
>
> To address this challenges, GPI-PINN 2 uses an expectation-free version $\mathcal{G}$ of the Hamiltonian $\mathcal{H}$ and leverages the $L^2$-optimal property of conditional expectation (Proposition 2.3) to learn the value function from $\mathcal{G}$, leading to the update rule (12). Combined with Proposition 2.2, this allows us to bypass the computation of the Hessian $\nabla^2_x V_{\theta}(t,x)$ and jump expectations $\mathbb{E}^{\mathcal{Z}}[V_\theta(t,x+\gamma(t,x,Z_1,a))]$, enabling faster numerical convergence for high-dimensional stochastic control problems with jumps. Moreover, GPI-PINN 2 only requires the backpropagation through the value network $\nabla_\theta V_\theta$ and thus to evaluate at most second-order derivatives, see Eq. (23).
>
>
> **Q2:**  *The experiments for consumption-investment do not compare the proposed algorithm with other standard DL frameworks as done for the LQR setting. Could the authors share these experimental results as well?*
>
> **Answer:** Yes, sure. Regardless of the specific dynamics considered in the control problems, our GPI-PINN algorithms consistently achieve results that are 1-2 orders of magnitude more accurate than existing model-free RL methods. Similarly, the local approaches mentioned in the introduction, which rely on a time-discretization of the state dynamics, only yield results for a restricted set of pre-specified points and fail to provide a global solution across the entire time-space domain, *i.e.* for any $(t,x) \in [0,T] \times D$.  Finally, additional experiments comparing the performance of GPI-PINN 2 with existing baseline DL algorithms for the consumption-investment problem are ready and can be shared in the revised version of the manuscript.
>
> **Q3:**  *Numerical experiments seem to only address dimensions d=10 to 50 and are limited to LQR and consumption-investment problem. It would be nice to mention larger scale problems where this could be applied.  Could the authors comment on the scalability and potential of this approach to larger scale environments? Are there environments or settings that are particularly suited to the strengths of the proposed algorithms?*
>
> **Answer:** Thanks for the suggestion. Of course, it depends on the computational resources. But we can easily apply GPI-PINN 2 to control problems with up to 100-150 dimensions. On the other hand, in our comparison study, GPI-PINN 1 does not scale beyond 10 dimensions with jumps and 50 dimensions without jumps, which confirms the efficiency of GPI-PINN 2 as outlined in Figure 1 (see also the computational complexity analysis provided to Reviewer 1, Q5). Since we wanted to compare GPI-PINN 1 and 2, we only reported results for both GPI-PINNs up to 50 dimensions. However, higher-dimensional results for GPI-PINN 2 are available and could be included in the paper to further illustrate the scalability of the approach.
>
>
> As long as the state dynamics of the control problem are known, GPI-PINN 2 offers an efficient and scalable approach that provides accurate solutions in high dimensions for both the value function and the optimal control at any point in the time-space domain $[0,T] \times D$, while accommodating jumps. From a pure practical standpoint, it performs particularly well on LQR-type of control problems, regardless of the jump-size distribution. Note that even when the systems dynamics are not known, nothing prevents us from learning these dynamics in a preliminary step using some recent model-learning (RL) algorithm on observed data, with *e.g.* Brunton et al. (2016) or Champion et al. (2019). Once an approximate model is identified, GPI-PINN 2 can then be applied to efficiently solve the resulting control problem.
>
> **Weakness 3:** We will improve the quality of the plots.
>
>  - Brunton, S. L., Proctor, J. L., & Kutz, J. N. (2016). Discovering governing equations from data by sparse identification of nonlinear dynamical systems. _Proceedings of the national academy of sciences_, _113_(15), 3932-3937.
> - Champion, K., Lusch, B., Kutz, J. N., & Brunton, S. L. (2019). Data-driven discovery of coordinates and governing equations. _Proceedings of the National Academy of Sciences_, _116_(45), 22445-22451.

---

> > ### Comment · Reviewer_9fvu · 2025-08-06
> > **Response**
> >
> > Thank you for your detailed responses.
> >
> > Re Q1:
> > Thank you for your elaboration. I think the paper would benefit from a description along these lines either in the introduction or after describing both algorithms inside a new subsection.
> >
> > Re Q2:
> > Thank you for the answer, this would alleviate my concerns, but unfortunately authors are not able to share figures or updates to the manuscript now. If it is allowed, I would be curious to see some summary results in a small table to evaluate the performance improvement over the specific baselines that the authors have now implemented.
> >
> > Re Q3:
> > Thank you. Yes, including larger scaling for GPI-PINN-2 is essential for demonstrating its strengths. In the current form, not including these results is a significant lack of evaluation. Similar to Q2, I would be interested in seeing summary results in a table if it is allowed.

---

> ### Author Response · Authors · 2025-08-07
>
> **Re Q1:** Indeed, we totally agree.
>
> **Re Q2:** Sure, here is the table showing the mean absolute error (MAE) with respect to the true value function  $V$ on a test set of size $M=10,000$ for the consumption-investment problem  with jumps of Section 4.2 (51 dimensions). The MAE for the value function is defined as $MAE_V = \frac{1}{M} \sum_{i=1}^M  | V_{\theta^{(k)}} (t_i, x_i) - V(t_i, x_i) |$, and is reported as a function of the number of epochs $k$.
> |  Epock $k$|  GPI-PINN2 | Han and E. |  PPO |  SAC |
> |---|---|---|---|---|
> | 0| 7.9686  | 7.5324  |  13.93 |  15.93 |
> | 50  |  2.2252 |  2.1282 |  9.0250 | 11.7632  |
> |  100 | 1.1140  | 1.6487  |  6.2568 |  8.5501 |
> | 500 |  0.0821 | 1.0143  |  4.1798 | 6.3672  |
> | 1000  |  0.0245 | 0.7345  |  2.9132 |  5.7546 |
> | 5000  | 0.0119  |  0.6872 |  2.5231 |  4.4817 |
>
> Note that these algorithms are configured in such a way to ensure equal computing time per epoch (in seconds). Similar results are also observed for the mean absolute error with respect to the true optimal control $\alpha^* $, i.e. $MAE_\alpha = \frac{1}{M} \sum_{i=1}^M \| \alpha_{\phi^{(k)}}(t_i, x_i) - \alpha^*(t_i, x_i) \|_2$. These findings are consistent with Figure 3 in the LQR setting and further confirm the superiority of our approach over standard baseline algorithms for high-dimensional stochastic control problems with jumps.
>
> **Re Q3:** You can find below a table summarizing higher-dimensional results for GPI-PINN 2 in the LQR example with controlled jump intensity (Section 4.1) after $k^* = 5000$ epochs.
> |  Dimensions $(d)$|  $MAE_V$ | $MAE_\alpha$ |  Loss $\mathscr{L}_1$ |  Loss $\mathscr{L}_2$ | Time (sec.) |
> |---|---|---|---|---|---
> | 5 | 0.0023 | 0.0041  | 0.0237  | -0.1332 | 6410
> | 10| 0.0025 | 0.0049 | 0.0314 |  -0.1972 | 8093
> | 50  | 0.0147 | 0.0065  |   0.1267 | -0.4206  |16,129
> |  100 | 0.0492 |  0.0054|  0.697 |  -0.4272 | 24,359
> | 150 | 0.0547 | 0.0069  |  1.0545 | -0.4384 | 33,120
>
>  This illustrates the scalability of the approach. Thank you for your valuable suggestions.

---

### Official Review · Reviewer_iDp5 · 2025-07-02

**Clarity:** 3
**Significance:** 3
**Originality:** 2
**Rating:** 5
**Confidence:** 2

**Summary:**

In this paper, past works in deep learning for continuous-time stochastic control are extended by utilizing physics-informed neural networks to reduce the computational burden of learning the optimal control and value function through generalized policy iteration. The first algorithm introduced, GPI-PINN-1, trains the value network to satisfy the Hamilton-Jacobi-Bellman equation by setting the loss function to be the residual. In this version the authors bypass computing the gradients and Hessian of the value function, but still need to evaluate the jump expectations in order to evaluate the Hamiltonian. GPI-PINN-2 not only bypasses the gradient computations of all orders, but also avoids explicitly evaluating the jump expectations by exploiting the known state dynamics, since the value target can be expressed in terms of these quantities. Furthermore, they use an iterative approach where the value network learned in the past iteration step is involved in computing the target for the new iteration step. However, this computational efficiency comes at the cost of reduced stability during training. The performance and training time of GPI-PINN-1 and 2 are compared, revealing that GPI-PINN-2 is by far the more practical algorithm when jumps must be accounted for despite its stability issues. Finally, numerical experiments on continuous-time stochastic control demonstrate the superiority of GPI-PINN-2 over existing model-free and model-based algorithms.

**Questions:**

**Q1:** How is the stopping condition in algorithm 1 evaluated efficiently?

**Q2:** How high-dimensional can the problem get before GPI-PINN-2 struggles to solve it?

**Q3:** What was the actual batch size per forward pass used when training the GPI-PINN networks? Was this batch size (and the number of batches per epoch) also used for the baseline deep learning algorithms?

**Ethical Concerns:**

["NO or VERY MINOR ethics concerns only"]

**Final Justification:**

Overall, I believe that the authors have responded to the concerns of myself and the other reviewers effectively. Hence, I am raising my score to a 5.

Issues resolved:
- An explanation of how the stopping condition is evaluated efficiently (using samples that provide coverage of the entire state space and all possible points in time) was given. If the maximum error over all sample points remains small over several rounds, it is reasonable to assume that the value network has learned a good approximation of the value over the entire state space.
- The authors are prepared to show experimental results backing up the claim that GPI-PINN-2 can handle problems with more dimensions than were already considered in the paper
- The authors confirmed that the comparison between their method and the baselines was fair in terms of the time used for training
- The authors have shown that they can provide additional theoretical justifications for their choices in their responses to reviewers fySF and cXcY

Furthermore, I do not consider the similarities between this paper and the paper by Duarte to be grounds for rejection.

**Limitations:**

yes

**Paper Formatting Concerns:**

No concerns

**Quality:**

4

**Strengths And Weaknesses:**

**Quality:** The techniques used in this paper have a solid theoretical foundation, and this is reflected in the strong empirical performance of GPI-PINN-1 and 2. The experiments run give a good idea of the performance benefits that GPI-PINN-2 can provide over GPI-PINN-1 and the baseline methods, validating that it would be worth adopting is many scenarios. Overall, I consider this to be high-quality work.

**Clarity:** The overall clarity of the paper is good, with the algorithms being presented cleanly and the figures mostly being easy to interpret. However, there is a mismatch between the legend of the left plot of Figure 4 (which says the blue curve is the boundary loss) and the caption (which says the orange curve is the boundary loss). Furthermore, with the large amount of notation which needs to be introduced in the first two sections, adding a table in the appendix explaining each symbol may be a good idea.

There are a few questions which I had about the practical implementation and performance of the algorithms, which are given in the questions section.

**Significance:** The problem formulation is very general and could be applied to many control problems, with the caveat that the system dynamics must be known. As the authors state, there are plenty of problems where certain dynamics can be modeled using known physical laws; however, in a real-life system the overall dynamics are likely to be influenced by less predictable phenomena as well. Furthermore, it is not obvious that a model trained in this way will be robust to noisy state measurements during deployment.

**Originality:** This is the first application of physics-informed neural networks to this problem that I am aware of. However, the methods used are closely related to those introduced in [1], including the use of the least-squares Monte-Carlo method to bypass computing the jump expectation.

[1] Duarte, V., Duarte, D., and Silva, D. H. Machine learning for continuous-time finance. The Review of Financial Studies, 37(11):3217–3271, 2024.

---

> ### Author Rebuttal · Authors · 2025-07-25
>
> **Q1:**  *How is the stopping condition in algorithm 1 evaluated efficiently?*
>
> **Answer:** Thank you, we should have explained this more clearly. Since our value function is parameterized by a neural network $V_\theta$, it is straightforward to evaluate the difference between $V_{\theta^{(k)}}(t,x)$ and $V_{\theta^{(k-1)}}(t,x)$ on a set of sample points $(t,x)$ from $[0,T] \times D$. In practice, we repeat *Step 1* (value updating) $N_1$ times and *Step 2* (policy updating) $N_2$ times before evaluating the stopping condition. An alternative efficient stopping rule would be to simply fix a maximum number of epochs $k^*$ (as used in the comparison Figure 3). This should indeed be emphasized more clearly. A more implementable version of GPI-PINN 1 will also be added to the Appendix for clarity.
>
> **Q2:**  *How high-dimensional can the problem get before GPI-PINN-2 struggles to solve it?*
>
> **Answer:** In all our numerical experiments, GPI-PINN 2 can easily handle control problems with up to 100-150 dimensions. On the other hand, in our comparison study, GPI-PINN 1 cannot scale beyond 10 dimensions with jumps and 50 dimensions without jumps, which confirms the efficiency of GPI-PINN 2 as outlined in Figure 1 (see also the computational complexity analysis provided to Reviewer 1, Q5). For this reason, the original manuscript only reported results for both GPI-PINNs up to 50 dimensions.  However, higher-dimensional results for GPI-PINN 2 are ready and can of course be included to further confirm the scalability of the approach.
>
> **Q3:**  *What was the actual batch size per forward pass used when training the GPI-PINN networks? Was this batch size (and the number of batches per epoch) also used for the baseline deep learning algorithms?*
>
> **Answer:** We used a batch size of 256 with a number of steps $N_1 = N_2 =64$ when training the GPI-PINNs, leading to a total batch size of 16,384 per epoch $k$ before each update.  The batch size in the other baseline algorithms was configured in such a way to ensure roughly equal computing time per iteration in each method, so that a similar comparison plot as Figure 3 would be obtained if we replaced the epochs by the actual running time in seconds (*i.e.* batch size of 512 with approximately 64 steps per epoch for PPO, SAC, Han and E). We will emphasize this more clearly in Appendix 6.3. when describing the neural network architecture.
>
> **Originality:** Our method is indeed related to the paper "Machine learning for continuous-time finance (2024)" by Duarte, Duarte and Silva. Their main focus is on lower-dimensional problems without jumps. But they also solve a 1-dimensional example with jumps. Our value function update rule is different, and we consider high-dimensional problems with finite time horizon. Since we have a finite time horizon, we also have a terminal condition, which we write as an additional term into the loss function.

---

> > ### Comment · Reviewer_iDp5 · 2025-08-03
> > **Reply to Rebuttal**
> >
> > Thank you for your response. I have no further comments.

---

### Official Review · Reviewer_cXcY · 2025-07-02

**Clarity:** 4
**Significance:** 3
**Originality:** 3
**Rating:** 3
**Confidence:** 4

**Summary:**

This paper investigates a finite-horizon continuous-time stochastic control problem with jumps. The associated Hamilton–Jacobi–Bellman (HJB) equation includes a nonlocal integral term due to the jumps, which makes the problem harder than normal optimal control problem. The authors propose a two-step iterative scheme: for a fixed feedback control, the value function is updated by minimizing the residual of the HJB equation; then, holding the value function fixed, the control is improved by minimizing the Hamiltonian. Two variants of the algorithm are introduced: the first uses multiple samples to approximate the jump expectation via Monte Carlo, while the second simplifies this step by using a single sample per spatial point. Numerical experiments are presented to demonstrate the performance and efficiency of both approaches. Overall, the paper is well-written.

**Questions:**

The design of Algorithm 2 raises two conceptual concerns that should be clarified. I would suggest the paper be accepted if the authors could address this issue.

In the first part, you modified the PINN loss by fixing the parameter $\theta$ in G and only optimize the term $V_\theta$ in line 191.5. (I temporally ignore the boundary trem.) Reorganizing the terms, the bracket before taking square in loss $\mathcal{L}_1$ can be viewed as

$$V_\theta – V_{\theta^{(k)}} – RES_{HJB}(\theta^{(k)}),$$

where $RES_{HJB}$ is the residual of the HJB equation (before taking expectation) in the PINN loss. This update means we hope that the new value function is close to the old one added by the HJB residual. However, since $E[RES_{HJB}] = 0$ characterizes the true solution, there's no justification for using the residual itself as an additive update. In fact, using

$$V_\theta – V_{\theta^{(k)}} + RES_{HJB}(\theta^{(k)}),$$

(the negative of the update) seems equally justified, leading to opposite update directions. The current construction appears arbitrary without additional justification.

For the second part, the Hamiltonian is expected to be maximized after taking expectation, while the algorithm only uses one sample for each x. Please explain why it is fine we only use one sample?

Below are some minor questions or comments.

1.	Page 2 line 71. Duan et al is not the original paper for TRPO. It is better to cite the original work.

2.	Page 3 line 104. HBJ should be HJB.

3.	Page 4 eq (11). One the left, should it be just a value function without the input?

4.	Page 5, part 2 of the first algorithm. I would suggest the authors add a remark that the term $\partial_t v$ in $\mathcal{H}$ does not involve I the optimization, so it is just minimizing the Hamiltonian. (At first glance, it seems you are updating the control using the residual of the HJB equation.)

5.	The algorithm that consecutively update the value function and the control is very similar to the actor-critic methos in reinforcement learning. I think papers that implement this method in optimal control method such as
Solving time-continuous stochastic optimal control problems: Algorithm design and convergence analysis of actor-critic flow
Can be added to related work.

**Ethical Concerns:**

["NO or VERY MINOR ethics concerns only"]

**Final Justification:**

The authors resolved most of my questions. My recommendation is not changed.

**Limitations:**

Yes

**Paper Formatting Concerns:**

Nan

**Quality:**

3

**Strengths And Weaknesses:**

Strength

The paper is well-written, with key concepts and algorithmic steps clearly explained.

The use of Proposition 2.2 enables an efficient approximation of the second-order differential operator without computing gradients and Hessians directly, which is beneficial for high-dimensional problems.

The second algorithm reduce computational costs.

The authors present detailed numerical experiments

Weakness

While the second algorithm reduces the computational cost, it lacks theoretical justification. I have put details in Questions.

---

> ### Author Rebuttal · Authors · 2025-07-25
>
> **Q1:** *In the first part, you modified the PINN loss by fixing the parameter  $\theta$ in $G$ and only optimize the term  $V_\theta$  in line 191.5. (I temporally ignore the boundary trem.) Reorganizing the terms, the bracket before taking square in loss  $\mathcal{L_1}$ can be viewed as*
> $$V_\theta - V_{\theta_k} - RES_{HJB}(\theta^{(k)})$$
>
> *where $RES_{HJB}$  is the residual of the HJB equation (before taking expectation) in the PINN loss. This update means we hope that the new value function is close to the old one added by the HJB residual. However, since  $\mathbb{E}[RES_{HJB}]$  characterizes the true solution, there's no justification for using the residual itself as an additive update. In fact, using*
> $$V_\theta - V_{\theta_k} + RES_{HJB}(\theta^{(k)})$$
>
> *(the negative of the update) seems equally justified, leading to opposite update directions. The current construction appears arbitrary without additional justification.*
>
>
> **Answer:** This is an excellent remark. More generally, using Proposition 2.3, we can always rewrite the bracket in $\mathcal{L}^1$ (l.191) as
> $$V_\theta - V_{\theta_k} - \pi_k RES_{HJB}(\theta^{(k)}) \ ,$$
>
> where $\pi_k \in \mathbb{R}$ is an adaptive scaling factor for the expectation-free HJB residuals that may depend on the epoch $k$.
>
>
> From a practical perspective, we empirically observe that the simple rule $\pi_k \approx 1$ provides a good trade-off between convergence speed and accuracy of the improvements in GPI-PINN 2. Exponentially decaying  $\pi_k > 0$ also perform well in practice (similarly to learning rate schedules), while we observe that negative scaling factors always fail to converge to the true solutions with exploding losses $\mathcal{L}^1$ and $\mathcal{L}^2$ after only a few epochs.
>
>
> Theoretically, this is related to the convergence properties of the GPI-PINN 2 algorithm, which are still under research. But roughly, the observed empirical behavior can be understood by considering the following extended operator at epoch $k$,
> $${\small\mathcal{T}^k V \hspace{-0.5mm}= \hspace{-0.5mm} V  \hspace{-0.5mm}+ \pi_k  \hspace{-0.5mm}\left[ RES_{HJB}(V)+\frac{h}{2} RES^2_{HJB}(V)+ \frac{h^2}{3} RES^3_{HJB}(V) + \cdots \right]} , $$
>
> such that $\theta^{(k+1)} = \mathcal{T}^k V_{\theta^{(k)}}$. This extended operator $\mathcal{T}^k$ can be shown to be contractive for specific values of $\pi_k > h >0$, while it is always expanding for $\pi_k <0$. For simplicity, we did not include this extension in the original manuscript, but we agree that incorporating these empirical observations along with detailed numerical results and a brief theoretical justification concerning the choice of $\pi_k$, could strengthen a revised version of the manuscript.
>
> **Q2:** *For the second part, the Hamiltonian is expected to be maximized after taking expectation, while the algorithm only uses one sample for each x. Please explain why it is fine we only use one sample?*
>
> **Answer:** Yes, correct. This is a bit like stochastic gradient descent. Every single sample point involves an error. But over many iterations, the errors average out. We see in our numerical experiments that we have more oscillations in the training of GPI-PINN 2 compared to GPI-PINN 1, but the loss still goes to zero. See Duarte et al. (2024), Duan et al. (2023) and Cohen et al. (2025) for theoretical convergence results supporting this approach.
>
> **Q3:** *Below are some minor questions or comments.*
>
> 1.  *Page 2 line 71. Duan et al is not the original paper for TRPO. It is better to cite the original work. Page 3 line 104. HBJ should be HJB.*
>
> Yes, thank you. We will refer to the original paper.
>
> 2.  *Page 4 eq (11). One the left, should it be just a value function without the input?*
>
> Thanks, but we need the inputs since (11) describes a non-linear regression of $\mathcal{G}$ on $X^\alpha_t$ at time $t$.
>
> 3.  *Page 5, part 2 of the first algorithm. I would suggest the authors add a remark that the term  $\partial_t v$  in  $H$  does not involve in the optimization, so it is just minimizing the Hamiltonian. (At first glance, it seems you are updating the control using the residual of the HJB equation.)*
>
> Indeed, you are right.
>
> 5.  *The algorithm that consecutively update the value function and the control is very similar to the actor-critic methos in reinforcement learning. I think papers that implement this method in optimal control method such as Solving time-continuous stochastic optimal control problems: Algorithm design and convergence analysis of actor-critic flow Can be added to related work.*
>
> Yes, you are right. We will add these references.
>
> Thank you for the reference, this will be added.
>
>
> - Cohen, S. N., Hebner, J., Jiang, D., & Sirignano, J. (2025). Neural Actor-Critic Methods for Hamilton-Jacobi-Bellman PDEs: Asymptotic Analysis and Numerical Studies. _arXiv preprint arXiv:2507.06428_.
> - Duan, J., Li, J., Ge, Q., Li, S. E., Bujarbaruah, M., Ma, F., & Zhang, D. (2023). Relaxed actor-critic with convergence guarantees for continuous-time optimal control of nonlinear systems. _IEEE Transactions on Intelligent Vehicles_, _8_(5), 3299-3311.
> - Duarte, V., Duarte, D., & Silva, D. H. (2024). Machine learning for continuous-time finance. _The Review of Financial Studies_, _37_(11), 3217-3271.

---

> > ### Comment · Reviewer_cXcY · 2025-08-02
> > **Thank you for the response**
> >
> > Thank you for the response. I am still confused about the update direction of HJB residual. I do not understand the Taylor expansion using the HJB residual. But I trust the authors that this method works well empirically.

---

> ### Author Response · Authors · 2025-08-04
>
> Thank you, we can add a more careful discussion of this point to our paper. Put simply, for $\pi_k < 0$, the updating operator of GPI-PINN 2 is expanding, which causes the loss $\mathcal{L}^1$ to diverge to infinity. In this case, GPI-PINN 2 does not converge. On the other hand, for $\pi_k > 0$, the updating operator of GPI-PINN 2 can be contracting. So, for suitably chosen step sizes, the iteration converges. This behavior is confirmed numerically.

---

> > ### Comment · Area_Chair_MZeR · 2025-08-05
> > **Please continue participating in the discussion**
> >
> > Dear reviewer,
> >
> > Is the reply of the authors satisfactory for you? I would like to encourage the reviewer and the authors to continue this disussion because this point seems to be critical about the paper. I would also appreciate other reviewers' involvement with this discussion.
> >
> > Thank you,
> > Area Chair

---

> > > ### Comment · Reviewer_cXcY · 2025-08-05
> > > **Thank you for the reply**
> > >
> > > To the authors:
> > >
> > > I appreciate the authors’ effort to address my concern. It appears that the theoretical justification for the proposed algorithm is still an open question and remains under investigation. I thank the authors for their response and will maintain my original score.
> > >
> > >
> > > To the area chair:
> > >
> > > While the authors’ explanation does not fully resolve the theoretical concern I raised, the rest of the paper is well-written. If the final decision is to accept the paper, I am supportive.

---

> > > > ### Comment · Area_Chair_MZeR · 2025-08-05
> > > >
> > > > To Reviewer: Thank you for your participation! I'd like to get a better understanding of the theoretical concern (I have not read the paper yet), and I appreciate your help.
> > > > Is the concern critical, in the sense that the justification of the method is wrong and a method is build on a wrong justifiication?
> > > > Or is it more like this is an open theoretical question that can be studied in a future work, and if it is not addressed in this paper, it is not deterimental to the integrity of the work.
> > > >
> > > > A caricature of the first case is that a method suggests performing gradient *ascent* to minimize a function.
> > > > A caricature of the second case is that the GD is suggested to minimize a function, but there is no convergence rate proof.
> > > > (Suppose that we are not in our current state of knowledge, but several decades ago.)
> > > >
> > > > Put it differently, we want to make sure an accepted paper at NeurIPS is a correct work, as much as we can be sure. Completeness is a secondary objective.
> > > >
> > > > Thank you,
> > > > Area Chair

---

> > > > > ### Comment · Reviewer_cXcY · 2025-08-05
> > > > > **Reply to area chair**
> > > > >
> > > > > Dear area chair,
> > > > >
> > > > > Thank you for your follow-up. The problem is $\textbf{not critical}$ because the authors already have a solid algorithm 1. I am questioning on algorithm 2, which aims to reduce the computational cost. Within algorithm 2, my concern lies in computing the value function V, which is usually referred to as “critic” in reinforcement learning.
> > > > >
> > > > > You provide a very good analog for my concern. The goal is to obtain the $\textbf{critical point}$ of a function $f(x)$, i.e. we hope $\nabla f(x) = 0$ (analog to satisfying $RES_{HJB}=0$). The authors propose an update method resembling gradient descent and show nice numerical results. However, my concern is: without further knowledge of $f$, how do we know that gradient descent (rather than ascent, or some other direction) will move us toward the correct critical point?
> > > > >
> > > > > The authors also give an explanation of this issue using an argument of series, but I am sorry that I do not fully understand this argument.

---

> > > > > > ### Comment · Area_Chair_MZeR · 2025-08-06
> > > > > >
> > > > > > Dear Reviewer,
> > > > > > Thank you very much for your explanation. This is very helpful for me to understand that state of the paper.

---

> > > > > > > ### Author Response · Authors · 2025-08-07
> > > > > > >
> > > > > > > Dear Area Chair, dear Reviewers,
> > > > > > >
> > > > > > > Thank you again for the interesting discussion. The main purpose of this submission has been the introduction of an efficient deep learning method for solving high-dimensional stochastic control problems with jumps and our numerical experiments yield good results in the examples we studied. It might be worth mentioning that we tried to provide in our previous replies a short intuitive theoretical justification for the GPI-PINN 2 updating rule based on a Taylor series argument. But prompted by your question about the correct sign of the gradient steps in this value function updating rule, we have developed a more compelling theoretical argument, based on a physics-type heuristic Taylor expansion and on contracting semigroups, which provides a range of scaling/learning rates $\pi_k$ for which our 2nd algorithm converges. These theoretical results are in line with the proposed methodology in the submitted paper ($\pi_k =1$)  and  exclude negative learning rates. This is also confirmed by numerous numerical experiments: the algorithm converges if the learning rate is chosen in the right range, while it quickly explodes otherwise. If you find it relevant, we can share these developments and experiments with you.

---

### Official Review · Reviewer_fySF · 2025-07-05

**Clarity:** 3
**Significance:** 2
**Originality:** 3
**Rating:** 4
**Confidence:** 3

**Summary:**

This paper introduces a model-based deep learning approach for solving finite-horizon continuous-time stochastic control problems that feature jumps. The proposed method iteratively trains two neural networks: one to represent the optimal policy and another to approximate the value function. Leveraging a continuous-time version of the dynamic programming principle, the authors derive two distinct training objectives based on the Hamilton-Jacobi-Bellman (HJB) equation, which aim to ensure the networks capture the underlying stochastic dynamics. They propose two algorithms, GPI-PINN 1 and GPI-PINN 2. GPI-PINN 1 minimizes HJB residuals, but is noted to be inefficient with jumps. GPI-PINN 2 is designed to circumvent costly computations of gradients, Hessians, and jump expectations, making it efficient for high-dimensional problems with jumps. Empirical evaluations on various problems, including linear-quadratic regulators and optimal consumption-investment problems (both with and without jumps), illustrate the accuracy and scalability of their approach, demonstrating its effectiveness in solving complex, high-dimensional stochastic control tasks.

**Questions:**

1.  In Algorithm 2, the function $\mathcal{G}$ plays a crucial role in updating the value and control networks. Could the authors provide a more intuitive explanation for how $\mathcal{G}$ relates to the original HJB equation and why its use helps circumvent the costly computations of expectations and higher-order derivatives? A deeper dive into the connection between minimizing $\mathcal{G}$ and satisfying the optimality conditions would enhance clarity.
2.  The paper highlights that model-free RL is less accurate when dynamics are known. Could the authors elaborate on how their approach (which relies on *known* dynamics) compares to recent advancements in model-learning within RL, where models are learned from data and then used for planning or control? This would provide a broader context for the "model-based" claim.
3. The paper mentions various hyperparameters (e.g., $M_1, M_2, k_*, \eta_1, \eta_2$, $\xi$, network architecture). A more detailed discussion on the sensitivity of the algorithms (especially GPI-PINN 2) to these parameters, and strategies for their robust selection across different problem instances, would be valuable for practical implementation. What are the criteria for selecting the proportionality factor $\xi$ (L209) and its impact on convergence?
4. The current formulation assumes specific characteristics for the random measure $N^\alpha$ and the distribution $Z$ of jump vectors. How general is the approach for different types of jump processes (e.g., non-Poissonian jumps, jumps with state-dependent or control-dependent distributions that are more complex than the assumed form)?
5.  While GPI-PINN 2 is stated to be more efficient, a more explicit computational complexity analysis (e.g., in terms of state dimension $d$, number of hidden layers $L$, and samples $M_1, M_2$) for both training and inference, compared to alternative methods, would solidify the claims about scalability.

**Ethical Concerns:**

["NO or VERY MINOR ethics concerns only"]

**Final Justification:**

Thanks for the clarification. However, I don't see this paper with "high impact on at least one sub-area of AI or moderate-to-high impact on more than one area of AI", so I will maintain my score to borderline accept this paper.

**Limitations:**

Yes.

**Paper Formatting Concerns:**

No major formatting concerns observed.

**Quality:**

3

**Strengths And Weaknesses:**

This paper presents a rigorous and detailed model-based deep learning approach to a challenging class of stochastic control problems. Its strengths lie in its theoretical foundation and the proposed methods' efficiency for high-dimensional problems with jumps. However, there are some areas where the clarity and comprehensiveness of the presentation could be improved.

**Strengths:**
*  The paper tackles continuous-time stochastic control problems with jumps in high dimensions, which is a complex and relevant area. The development of GPI-PINN 2, specifically designed to handle jumps efficiently by avoiding computationally expensive gradient, Hessian, and expectation computations, is a notable contribution.
* Unlike many model-free deep RL algorithms that rely solely on environmental sampling, this approach explicitly leverages known underlying dynamics (HJB equation). This design choice is claimed to lead to higher accuracy when state dynamics are fully known.
* The iterative training of two neural networks for both the value function and optimal control provides a global approximation across the entire space-time domain, allowing for rapid online evaluation. This is an advantage over local approaches.
*  The method's effectiveness, scalability, and accuracy are demonstrated across various numerical examples. These include LQR problems with and without controlled jump intensities, and complex optimal consumption-investment problems.
* The approach is well-grounded in stochastic control theory, leveraging the HJB equation and providing propositions (2.2 and 2.3) that enable computational efficiencies.

**Weaknesses:**
* While propositions are provided, the transition from theoretical statements to the practical implementation within the algorithms (e.g., the exact form of $\mathcal{G}$ in Algorithm 2's update rules) could be clearer in the main text. Some key definitions from standard literature are referenced but full clarity requires consulting external sources.
* While the paper states that model-free RL algorithms are "less accurate in cases where the state dynamics (2) are fully known and given" and shows GPI-PINN 2 outperforming PPO and SAC in Figure 3, the explanation for why this is the case could be more deeply explored beyond "rely solely on sampling". A more nuanced discussion of the trade-offs (e.g., data efficiency, robustness to misspecification) would be beneficial.
* The paper identifies that a limitation is the need to know the underlying dynamics of the state process, which is "not always available in real-world applications". While this is acknowledged, the implications for practical deployment are significant and warrant further discussion on how this model-based approach would integrate with real-world scenarios where dynamics need to be learned or estimated.
* GPI-PINN 1's inefficiency with jumps is a major drawback. While GPI-PINN 2 addresses this, the practical utility of GPI-PINN 1 seems limited, and perhaps it should be framed more as a stepping stone to GPI-PINN 2 rather than a fully viable alternative.

---

> ### Author Rebuttal · Authors · 2025-07-25
>
> **Q1:** *In Algorithm 2, the function $\mathcal{G}$  plays a crucial role in updating the value and control networks. Could the authors provide a more intuitive explanation for how  $\mathcal{G}$  relates to the original HJB equation and why its use helps circumvent the costly computations of expectations and higher-order derivatives? A deeper dive into the connection between minimizing $\mathcal{G}$  and satisfying the optimality conditions would enhance clarity.*
>
> **Answer:** Thanks for asking. We will try to explain this better. A typical Hamiltonian contains the gradient and the Hessian of the value function. This becomes computationally prohibitive in high dimensions (say 20 or more). Moreover, if the control problem contains jumps, the Hamiltonian includes jump-expectations, which in general, have to be computed with nested Monte Carlo. This increases the computational effort further.
>
> $\mathcal{G}$ is an expectation-free version of the Hamiltonian, which bypasses the computation of the Hessian and jump-expectations and which can be interpreted as the expectation-free HJB residuals. Proposition 2.3 then uses the $L^2$-optimality property of conditional expectations to show that the value function $V^\alpha$ can be learned from $\mathcal{G}$ using the update rule (12) .
>
> Moreover, the GPI-PINN 2 update rule only requires the computation of $\nabla_\theta V_\theta(t,x)$, eliminating the need for backpropagation through the expectation-free Hamiltonian $\mathcal{G}$, which is evaluated using the previous weights $\theta^{(k-1)}$. While GPI-PINN 1 gives good results for low-dimensional problems without jumps, GPI-PINN 2 is much more efficient for high-dimensional problems with jumps; see also our answer to Q1 of Reviewer 4.
>
> **Q2:** *The paper highlights that model-free RL is less accurate when dynamics are known. Could the authors elaborate on how their approach (which relies on known dynamics) compares to recent advancements in model-learning within RL, where models are learned from data and then used for planning or control? This would provide a broader context for the "model-based" claim.*
>
> **Answer:** Thanks, this is a good remark. Our approach indeed requires that the dynamics of the underlying system are known. But nothing prevents us from learning these dynamics in a preliminary step from data using a recent model-learning (RL) algorithm such as Brunton et al. (2016) or Champion et al. (2019). Once an approximate model has been learned, GPI-PINN  can then be applied to efficiently solve the resulting control problem. Including this perspective in our paper is a good idea. It will provide valuable context and broaden its scope.
>
> **Q3:** The paper mentions various hyperparameters (e.g., $M_1, M_2, \eta_1, \eta_2, \xi$, network architecture). A more detailed discussion on the sensitivity of the algorithms (especially GPI-PINN 2) to these parameters, and strategies for their robust selection across different problem instances, would be valuable for practical implementation. What are the criteria for selecting the proportionality factor*  $\xi$ (L.209) and its impact on convergence?
>
> **Answer:** Yes, you are right. Figures 1 and 11 show the sensitivity of our algorithms to the number of epochs $k^*$. Additional results should also be included for our grid search over $M_1, M_2, \eta_1, \eta_2$, although we observed that our algorithms' performance is robust with respect to these parameters (for $M_1, M_2$ sufficiently large). The most crucial hyperparameter for GPI-PINNs convergence is $\xi$, which we determined with the approach of Wang et al. (2022). They have shown that in the approximations of solutions of PDEs with infinite-width network (in the limit), the convergence rate of the PINN training error depends on the singular values of the neural tangent kernel associated with the PINN (which they derive explicitly). By choosing the proportionality parameters $\xi$ to be proportional to these singular values, they demonstrate empirically that the numerical convergence of PINNs and related algorithms can be significantly improved. We will emphasize this more clearly in Appendix 6.3, where the neural network architecture is described.
>
> **Q4:** *The current formulation assumes specific characteristics for the random measure  $N^\alpha$ and the distribution $Z$ of jump vectors. How general is the approach for different types of jump processes (e.g., non-Poissonian jumps, jumps with state-dependent or control-dependent distributions that are more complex than the assumed form)?*
>
> **Answer:** Our framework is very general. It covers non-Poissonian jump arrivals. Both, the jump intensities and jump-size distributions can depend on the time, state and control. The only thing it does not cover is infinite jump actvity, i.e. infinitely many small jumps during compact time-intervals. But such processes are a theoretical construct, which numerically, have to be approximated, typically by ignoring all jumps below a given threshold.
>
> **Q5:** *While GPI-PINN 2 is stated to be more efficient, a more explicit computational complexity analysis (e.g., in terms of state dimension $d$, number of hidden layers $L$, and samples $M_1$, $M_2$) for both training and inference, compared to alternative methods, would solidify the claims about scalability.*
>
> **Answer:** Thank you for the suggestion. A complete theoretical analysis is difficult. But we have some partial answers, and our numerical experiments strongly suggest that GPI-PINN 2 scales well.
>
> At inference, the computational complexity for evaluating the value network $V_\theta$ with depth $L$ and width $W$ is the same for GPI-PINN 1 and 2, namely
> $$\mathcal{O}(M_1 LW^2).$$
>
> Regarding training time, we have for GPI-PINN 1,
> $$\mathcal{O}(M_1 M_3 d^2LW^2+ M_2 LW^2) = \mathcal{O}(M_1 M_3 d^2LW^2),$$
>
> and for GPI-PINN 2,
> $$\mathcal{O}(M_1 dLW^2 + M_2 LW^2) = \mathcal{O}(M_1 dLW^2),$$
> where $M_3$ is the number of samples used to approximate the jump expectation $\mathbb{E}^{\mathcal{Z}}$ in GPI-PINN 1. This highlights the significant reduction in computational cost achieved by GPI-PINN 2 for training the value network $V_\theta$ in high-dimensional control problems with jumps, as confirmed by Figure 1. For the control network  $\alpha_\phi$, the training phase (Step 2) is the same between the two algorithms. Additional comparisons with existing deep learning algorithms could also be provided.
>
> - Brunton, S. L., Proctor, J. L., & Kutz, J. N. (2016). Discovering governing equations from data by sparse identification of nonlinear dynamical systems. _Proceedings of the national academy of sciences_, _113_(15), 3932-3937.
> - Champion, K., Lusch, B., Kutz, J. N., & Brunton, S. L. (2019). Data-driven discovery of coordinates and governing equations. _Proceedings of the National Academy of Sciences_, _116_(45), 22445-22451.
> - Wang, S., Yu, X., & Perdikaris, P. (2022). When and why PINNs fail to train: A neural tangent kernel perspective. _Journal of Computational Physics_, _449_, 110768.

---

> > ### Comment · Reviewer_fySF · 2025-08-08
> > **borderline accept**
> >
> > Thanks for the clarification. However, I don't see this paper with "high impact on at least one sub-area of AI or moderate-to-high impact on more than one area of AI", so I will maintain my score.

---

### Decision · Program_Chairs · 2025-09-17

**Decision:**

Accept (poster)

**Comment:**

This paper proposes a model-based actor-critic algorithm for solving a class of continuous-time stochastic control problem. The focus is on finite horizon problems. The method allows the dynamics to have jumps. The method benefits from the known dynamics to define PINN-style loss functions.

The paper introduces two methods: GPI-PINN 1 and GPI-PINN 2. The first method is not very efficient in dealing with jumps, while the second method is. The paper is theoretically motivated and shows the empirical effectiveness of the proposal methods.

The reviewers are mostly positive about the paper: we have one accept (5; confidence: 2), two borderline accepts (4; confidence: 3), and one borderline reject (3; confidence: 4).

On the positive side, the reviewers commented on the well-roundedness of the method in stochastic control theory, the generality of the method, empirical effectiveness, and that the paper is well-written, though sometimes dense in mathematical notation.
The major negative issue is that one step of the GPI-PINN 2 algorithm is not well-justified, as noted by Reviewer cXcY. There have been some discussions between the authors and the reviewer, but the issue is not fully resolved. The authors have a final message that they found a better theoretical justification for that part, but that was not explored further.

Without this justification, the paper basically offers two algorithms: one with all steps theoretically justified and another that is less theoretically justified in one of its steps. For both algorithms, we have empirical results showing their effectiveness. I confirmed with Reviewer cXcY that this issue was not critical. Overall, I think it is OK not to consider this issue a deal breaker.

I read the paper myself and I think this is a good paper. So, overall, I am recommending its acceptance.


I have one question: Equation (11) consider g to belong to the space of Borel measurable functions. This space is much larger than the space representable by NNs. This means that we may not obtain the minimum of the function. What is the effect of this on Proposition 2.3 and how the trick used based on it?